# Parameter Efficient Graph Encoding for Large Language Models

## Abstract

How can we best encode structured data into sequential form for use in large language models (LLMs)? In this work, we introduce a parameter-efficient method to explicitly represent structured data for LLMs. Our method, GraphToken, learns an encoding function to extend prompts with explicit structured information. The encoding function in GraphToken uses graph neural networks to effectively transfer the relational inductive biases in the structured data to an LLM. Unlike other work which focuses on limited domains (*e.g.*, knowledge graph representation), our work is the first effort focused on the general encoding of structured data to be used for various reasoning tasks. We show that explicitly representing the graph structure allows significant improvements to graph reasoning tasks. Specifically, we see across the board improvements - up to 73% points - on a wide variety of node, edge and, graph-level tasks on benchmarks for graph reasoning (GraphQA) and molecular property prediction tasks (ChemLLMBench).

## 1 Introduction

There has been an explosion of recent excitement around using LLMs (Vaswani et al., 2017; Devlin et al., 2018; Raffel et al., 2020; Brown et al., 2020; Touvron et al., 2023; Team et al., 2023) to represent, process, and analyze textual data. These models typically take sequential text as their input but recent work has extended inputs to spatial and temporal modalities (*e.g.*, to images as in Chen et al. (2022b) and to videos as in Arnab et al. (2021)).

Despite their success, current realizations of LLMs have noticeable problems – including a tendency to generate outputs which are untrue or unsupported by their prompt, commonly referred to as *hallucinations* (Wang et al., 2023a). Another intimately related issue is the problem of *freshness*, where the knowledge required to answer a query exists only after an LLM's training date (Vu et al., 2023). One mitigation for these problems is through the enrichment of the prompt with additional factual and fresh data. As Kadavath et al. (2022) showed, when LLMs are supplied with new and supporting information, they are capable of adapting their parametric beliefs to effectively incorporate new evidence. Despite the promise of these approaches to improve generation, most work in the area so far has focused primarily on the discovery and inclusion of relevant *textual* information (Khandelwal et al., 2019; Guu et al., 2020). However beyond text, there is an abundance of more *structured data* in many applications for LLMs. For example, structured data sources such as social and biological networks, chemical compounds, relational databases, rich tables and knowledge graphs are ubiquitous in industry. This begs the question - "How can we best include structured data in a LLM's context?"

Despite its importance, understanding how to best represent graph data optimally for LLM integration is an unsolved problem. Currently, the predominant mode of encoding structured data for LLMs is to use various types of *hand-crafted*, text-based serialization (Guo et al., 2023a; Wang et al., 2023b; Stechly et al., 2023) This approach can impose significant decoding complexity for the language model: from any text description, the model must first correctly decode and understand the structure before it can utilize the information. Recently, Fatemi et al. (2024) et al., demonstrated that pure text representations of structured data are insufficient for graph reasoning with LLMs. They show that LLMs are not able to utilize structure efficiently when posed with common reasoning tasks that are easily answered by classical graph algorithms. This highlights the need to explore better and more expressive ways of representing structured data to a LLM.

In this work, we propose GraphToken (Figure 1), a parameter-efficient method for encoding structured data for LLMs to address this deficiency. Inspired by recent advancements in parameter-efficient fine-tuning (Lester et al., 2021; Xu et al., 2023), GraphToken learns an encoding function that generates fine-tuned soft-token prompts. The soft-token prompt extends a textual prompt with explicit structural information, allowing us to train a much smaller number of GraphToken parameters compared to the total LLM parameter allocation. To enable the model to account for the relational inductive biases in the input, we employ various forms of graph neural networks within GraphToken. Unlike other options available for increasing graph reasoning performance (e.g. changing the model's pre-training mixture or fine-tuning the model's parameters Hu et al. (2021)), GraphToken's GNN adapter works with a frozen LLM, operating in the token-space of the model.

Our work is the first to develop parameter-efficient encoders specifically for general reasoning tasks on structured data. Our experimental results demonstrate that explicitly representing structure leads to significant improvement on the comprehensive GraphQA benchmark (Fatemi et al., 2024). For example, we show that adding a small number of "graph-aware" parameters can allow a large model like PaLM-2 S to outperform its much larger sibling PaLM-2 L by up to 41% accuracy.

Specifically, our contributions are as follows:

- **GraphToken**, a novel parameter-efficient encoder for structured data inclusion in LLMs.
- **Experiments**: Extensive experiments with a variety of both large proprietary and open source models that illustrate the benefits of GraphToken. Our experiments on both graph reasoning tasks and molecular property prediction show that our method can significantly improve LLM capabilities, allowing small models to outperform larger ones.
- **Analysis**: We analyze GraphToken generalization on both unseen tasks and graphs.

## 2 BACKGROUND

We introduce the related work in LLMs, prompting methods, Graph Neural Networks (GNNs), graph encoders, and graph models combined with LLMs.

### 2.1 LARGE LANGUAGE MODELS

**Pre-Trained Large Language Models (LLMs):** Language models (Rosenfeld, 2000; Zhao et al., 2023) are autoregressive models, assigning likelihoods to tokens given a context of preceding or surrounding tokens. While earlier methods counted frequencies of N-grams (Jurafsky, 2021, chapter 2), newer models adopted neural approaches with the advent of distributed word representations (Bengio et al., 2000; Mikolov et al., 2013). The increased power offered by the neural language models and the increase in model and dataset sizes has led to a new learning paradigm where large language models (LLMs) are pre-trained in an unsupervised setting on massive amounts of textual data and are used as base (foundation) models (Devlin et al., 2018; Radford et al., 2019). For each downstream application, the base model is fine-tuned on small amounts of task-specific data to adapt the model to the task.

**Parameter-Efficient Fine-Tuning:** With the rapid growth in the number of parameters for state-of-the-art LLMs (Achiam et al., 2023; Team et al., 2023) fine-tuning for each downstream task has become prohibitively expensive in both time and resources. The goal of parameter-efficient fine-tuning (PEFT) (Xu et al., 2023) is to adapt models to new tasks by updating only a small number of (possibly new) parameters. PEFT approaches are distinguished primarily by where parameters are tuned (or added). *Adapter-based approaches* (Houlsby et al., 2019; He et al., 2021) hold the LLM parameters frozen and add new trainable parameters to various parts of the model. Meanwhile, partial fine tuning and partial masking methods (Zhao et al., 2020; Zaken et al., 2021) only fine-tune or mask a subset of the LLM parameters – no new parameters are introduced. Some methods operate with frozen LLM parameters. *Soft-prompt* methods (Li & Liang, 2021a; Lester et al., 2021; He et al., 2022) prepend tokens with learnable parameters to the beginning of the LLM input or to the beginning of every LLM layer. Finally, the most popular group of PEFT techniques, *LoRA* and derivative methods (Hu et al., 2021; Edalati et al., 2022; Valipour et al., 2022) learn offsets to frozen model weights.

The closest related methods to this paper are the family of soft-prompt approaches. Most relevant is the work of Levine et al. (2022), where the input is provided to a separate trainable neural network to

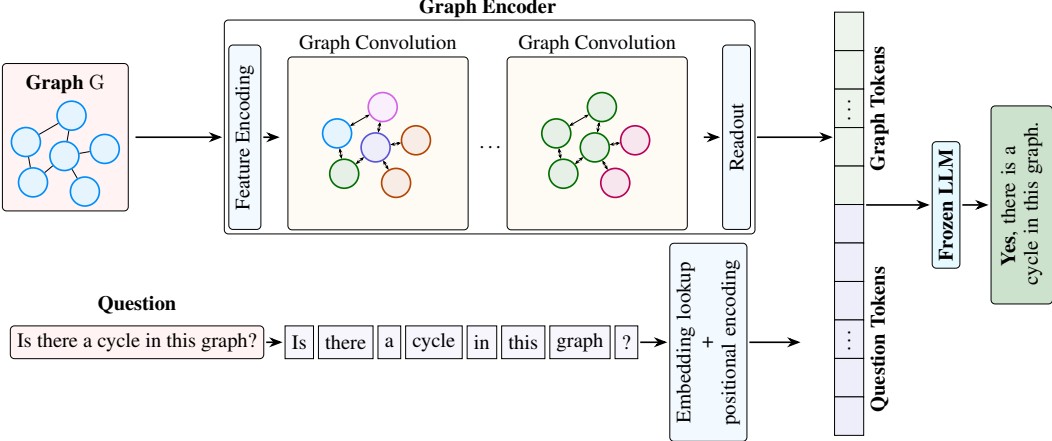

Figure 1: A visual overview of the architecture of GraphToken. The framework takes a graph and a corresponding question as input. The graph encoder takes the graph and generates graph tokens. The question is tokenized and embedded to question tokens. A frozen LLM leverages the graph and question tokens to generate an answer.

produce the soft-prompt. We extend this to encoding structured data input via a GNN to produce the LLM soft-prompt. In this work, we focus on approaches that keep the weights in the model frozen for two reasons. First, these methods are complimentary to the enormous body of ongoing work to improve model capabilities. We can easily use GraphToken with newer and more powerful models without imposing any architectural changes. Second, this class of PEFT approaches doesn't require many resources to train unlike methods which require full fine tuning, or many training examples.

## 2.2 GRAPH ENCODING WITH NEURAL NETWORKS

The field of graph representation learning (Chami et al., 2022) seeks ways to represent discrete structured data in a continuous domain, typically for use in some downstream learning task. While seminal work like DeepWalk (Perozzi et al., 2014) popularized the *node embedding* problem, later work utilized GNNs to generalize and learn representations of entire graphs (*graph embeddings*). Many approaches learning graph representations (*node* or *graph* embeddings) have followed since (Tsitsulin et al., 2018; Xie et al., 2022).

## 2.3 GRAPHS AND LLMS

The confluence of graph representation learning and reasoning with LLMs is a rapidly growing field of research – like language, structured data surrounds us but, unlike LLM input, it is not sequential. Some of the first graphs represented in the literature were knowledge graphs as in Agarwal et al. (2020), where the retrieval corpus of a retrieval LLM is augmented with text-encoded knowledge graphs. Ye et al. (2023) utilize instruction fine-tuned LLMs for node classification. Similarly, Chen et al. (2023b) leverage LLMs to enhance graph learning models by incorporating rich text attributes. Wang et al. (2023b) showed that language models demonstrate preliminary abilities for graph reasoning tasks. Later, Fatemi et al. (2024) proposed GraphQA – a comprehensive benchmark to systematically evaluate models for graph reasoning tasks – finding that graph reasoning tasks are currently difficult and that scaling laws do not seem to apply. Finally, in concurrent work to ours, Chai et al. (2023) study how prefix tuning Li & Liang (2021b) can extend graph reasoning capabilities. Interestingly their results are substantially different than ours. First, they were only able to operate on graphs where each node has rich textual features. With GraphToken we show that it is possible to use abstract graph representations (where the only node features are based on the graph itself (or are learned). Secondly, their proposed method (prefix tuning) is much more expensive than ours (prompt tuning) as adds additional parameters for each layer in the network - they require $\sim 10\times$ as many parameters as our method.

## 3 GRAPHTOKEN: STRUCTURALLY INFORMED GENERATION

When considering how to pass graph data to an LLM there are largely two families of options: (1) encoding it as lexical tokens for LLM embedding as in Fatemi et al. (2024) or (2) encoding it directly as a continuous representation via a neural network – skipping any LLM token embedding. While representing a graph as a sequence of lexical tokens has benefits in terms of interpretability, it introduces a core problem – there is often no clear choice in what order to sequentially write the graph data. In fact, this seemingly simple act of choosing a sequential serialization removes the permutation equivariance which is a core inductive bias for geometric machine learning (Bronstein et al., 2017). We believe a text encoding of structured data prohibits rich, concise, and expressive representations. Instead, our method eschews representing a graph in text in favor of directly producing – using a GNN as an encoder – the continuous representations for the LLM input. Our approach allows the GNN to provide sample efficient graph learning (and when necessary, other properties like permutation equivariance) for graph algorithmic tasks (Sanford et al., 2024). We refer to these new graph encoder learned soft-tokens in the LLM embedding space as "graph tokens."

To maintain the reasoning and language capabilities of the LLM, we freeze its parameters and teach the graph encoder to align its output representations with the LLM embedding space. In other words, we learn only the parameters of the graph encoder during the training process. This reduces computational requirements significantly (graph encoder parameters are minuscule compared to the LLM). During our tests, the LLM is prompted with the output of the graph encoder and a task about the graph, for example: *'Does this graph contain a cycle?'*. As such, the quality of the results is purely a function of how well the graph encoder represents the answer to the task and how well the LLM interprets that output.

### 3.1 ARCHITECTURE

An overview of the architecture is provided in Figure 1. At a high level, our model only has two components. First, the graph encoder takes a graph as input and outputs a fixed number of token embeddings. These tokens are then prepended to the sequence of initial token embeddings in the prompt for an LLM, which is then decoded to produce an answer as text.

**Graph Encoder**. GNN models range from simple averaging methods to complex models with multi-headed attention. Thus there are a wide variety of graph representations possible. We suspect that some of these representations are more suited to be consumed by an LLM. Therefore, we conducted a thorough study that includes several well-known graph encoder choices in §4.3. Our graph encoder takes the relational structure of the graph as input, using some form of graph positional encoding as node features (either learned, Laplacian, or a combination thereof) - see §4.3.2 for details.) Next, we apply a GNN to obtain a representation of the graph, which we read out using one of a few different techniques techniques depending on the task.

- For **graph-level** tasks (*e.g.*, `cycle check`) we do global pooling for readout, taking the *mean* or *sum* of the representations over all of the nodes.
- For **node-level** tasks (*e.g.*, `node degree`) we separately output the representation of each node. This can be optionally row-wise concatenated with a graph-level pooling.
- For **edge-level** tasks (*e.g.*, `edge existence`), we use a global representation or the two node-level representations concatenated.

We note that the readout used (*e.g.*, mean or sum pooling) is a hyper-parameter chosen during model selection. Whichever the readout technique, this representation is then projected onto the token space used by the LLM with a final dense layer.

**LLM**. For the experiments in the paper we use several LLMs, including PaLM 2 (Anil et al., 2023), Gemma2 (Team et al., 2024), Mistral7B (Jiang et al., 2023), and Phi2 (Javaheripi et al., 2023). however, our method generalizes to nearly any LLM in use today. For our purposes, any language model which can accept a sequence of token embeddings and produce text is acceptable, so long as it is possible to compute a gradient with respect to part of the input sequence. Some details about training are available in §7.

Table 1: Results comparing GraphToken against prompt engineering and soft prompting on graph reasoning tasks using the GraphQA$_{\text{Test}}$ benchmark (Fatemi et al., 2024), by accuracy. We see that GraphToken substantially improves PaLM 2-S performance on all graph, node, and edge-level tasks.

| Method | Graph Tasks | | | | Node Tasks | | Edge Tasks | | |
|---|---|---|---|---|---|---|---|---|---|
| | Node count | Edge count | Cycle check | Triangle count | Node degree | Connected nodes | Reach-ability | Edge existence | Shortest path |
| PALM 62B ZERO-SHOT | 0.217 | 0.124 | 0.760 | 0.015 | 0.140 | 0.147 | 0.849 | 0.445 | 0.115 |
| PALM 62B ZERO-COT | 0.146 | 0.094 | 0.323 | 0.127 | 0.104 | 0.088 | 0.735 | 0.335 | 0.336 |
| PALM 62B FEW-SHOT | 0.253 | 0.120 | 0.374 | 0.030 | 0.174 | 0.124 | 0.794 | 0.368 | 0.227 |
| PALM 62B COT | 0.276 | 0.128 | 0.580 | 0.081 | 0.292 | 0.131 | 0.452 | 0.428 | 0.386 |
| PALM 62B COT-BAG | 0.269 | 0.125 | 0.521 | 0.081 | 0.280 | 0.158 | 0.452 | 0.373 | 0.404 |
| PALM 2-S ZERO-SHOT | 0.365 | 0.121 | 0.747 | 0.006 | 0.414 | 0.250 | 0.835 | 0.482 | 0.020 |
| PALM 2-S ZERO-COT | 0.313 | 0.131 | 0.165 | 0.005 | 0.074 | 0.147 | 0.837 | 0.370 | 0.010 |
| PALM 2-S FEW-SHOT | 0.400 | 0.169 | 0.404 | 0.011 | 0.369 | 0.229 | 0.811 | 0.475 | 0.028 |
| PALM 2-S COT | 0.417 | 0.194 | 0.425 | 0.014 | 0.443 | 0.230 | 0.811 | 0.576 | 0.037 |
| PALM 2-S COT-BAG | 0.444 | 0.208 | 0.396 | 0.014 | 0.437 | 0.227 | 0.823 | 0.552 | 0.037 |
| PALM 2-S PROMPT TUNING | 0.056 | 0.018 | 0.832 | 0.162 | 0.098 | 0.068 | 0.838 | 0.544 | 0.462 |
| **PaLM 2-S GraphToken** (ours) | **0.996** | **0.426** | **0.956** | **0.348** | **0.962** | **0.264** | **0.932** | **0.738** | **0.638** |
| %Gain vs. PALM 2-S ZS | +172.9% | +124.2% | +28.0% | +5700% | +132.4% | +5.60% | +11.6% | +53.1% | +3090% |

Comparing this smaller LLM (PaLM-2-S) with GraphToken to its large equivalent (PaLM 2-L), the larger model now underperforms the smaller LLM in most graph reasoning settings:

| Method | | | | | | | | | |
|---|---|---|---|---|---|---|---|---|---|
| PALM 2-L ZERO-SHOT | 0.763 | 0.306 | 0.833 | 0.121 | 0.551 | 0.195 | 0.841 | 0.475 | 0.628 |
| PALM 2-L ZERO-COT | 0.739 | 0.244 | 0.224 | 0.037 | 0.079 | 0.395 | 0.710 | 0.416 | 0.230 |
| PALM 2-L FEW-SHOT | 0.603 | 0.359 | 0.738 | 0.214 | 0.558 | 0.461 | 0.851 | 0.415 | 0.607 |
| PALM 2-L COT | 0.622 | 0.344 | 0.727 | 0.223 | 0.597 | 0.452 | 0.815 | 0.522 | 0.687 |
| PALM 2-L COT-BAG | 0.631 | 0.346 | 0.692 | 0.220 | 0.600 | 0.450 | 0.846 | 0.604 | 0.680 |

## 4 EXPERIMENTS

In this section, we summarize the key experiments conducted with GraphToken. We begin by highlighting some of the most exciting results from our analysis here:

- **R1**: GraphToken demonstrates superior performance compared to established baselines across a comprehensive range of graph reasoning and molecular property prediction tasks.
- **R2**: GraphToken can significantly increase the capabilities of smaller models. In both graph reasoning and molecule property prediction tasks, we see GraphToken drastically boosting performance of smaller models, making them competitive with much larger models on most tasks.
- **R3**: GraphTokens can generalize to both unseen graphs and unseen graph tasks.

**Datasets.** We conduct our experiments on the graph reasoning tasks proposed in GraphQA (Fatemi et al., 2024). This dataset presents multiple graph reasoning problems with different difficulty levels. These tasks can be categorized as follows.

- **Graph-level.** `node count` (counting the number of nodes in a graph), `edge count` (counting the number of edges in a graph), `cycle check` (determining whether a graph contains a cycle), and `triangle count` (counting the number of triangles in a graph).
- **Node-level.** `node degree` (calculating the degree of a given node in a graph), and `connected nodes` (finding all the nodes that are connected to a given node in a graph),
- **Edge-level.** `reachability` (finding if there is a path from one node to another), `edge existence` (whether a given edge exists in a graph, and `shortest path` (finding the length of the shortest path from one node to another).

**Setting.** Details about the setting and reproducibility are available in §7. [1]

### 4.1 EXPERIMENT 1: GRAPHTOKEN PERFORMANCE

In this experiment, we rigorously evaluate the performance of GraphToken against the following comprehensive set of established baselines (described for brevity's sake in §A.3).

---

[1]To accelerate future research, we will open-source our code upon acceptance of the paper.

Table 2: Results comparing individual graph encoder performance with GraphToken and PaLM 2-S on GraphQA$_{Test}$ tasks, by accuracy. Note that there is 'no free lunch' here – no single encoder examined dominates across all tasks. Best result is bolded, second-best is underlined.

| | Method | Graph Tasks | | | | Node Tasks | | Edge Tasks | | |
| | | Node count | Edge count | Cycle check | Triangle count | Node degree | Connected nodes | Reachability | Edge existence | Shortest path |
|---|---|---|---|---|---|---|---|---|---|---|
| Non-linear | GCN | 0.746 | 0.056 | **0.964** | 0.208 | 0.264 | **0.264** | 0.918 | 0.680 | 0.604 |
| | GIN | 0.704 | 0.052 | 0.898 | 0.194 | 0.252 | 0.180 | 0.902 | 0.650 | 0.586 |
| | MPNN | 0.792 | 0.368 | 0.956 | **0.348** | **0.962** | 0.250 | 0.934 | 0.648 | **0.638** |
| | HGT | 0.252 | 0.084 | 0.934 | 0.234 | 0.266 | 0.184 | **0.944** | 0.718 | 0.600 |
| | MHA | 0.912 | 0.264 | 0.962 | 0.266 | 0.552 | 0.244 | 0.932 | **0.738** | 0.608 |
| Linear | Node Set | **0.996** | 0.080 | 0.948 | 0.198 | 0.190 | 0.118 | 0.942 | 0.596 | 0.568 |
| | Edge Set | 0.618 | **0.426** | **0.964** | 0.228 | 0.220 | 0.096 | 0.904 | 0.592 | 0.568 |

Table 3: Molecular property prediction benchmarks, measured by accuracy. GraphToken allows Gemma-2 2B to outperform larger models, even with parameter-efficient finetuning methods.

| | Trainable parameters | BACE | BBBP | ClinTox |
|---|---|---|---|---|
| Majority class | - | 0.640 | 0.700 | 0.808 |
| Gemma-2B (zero-shot) | - | 0.000 | 0.700 | 0.323 |
| Phi2-2.7B (zero-shot) | - | 0.000 | 0.000 | 0.010 |
| Mistral-7B (zero-shot) | - | 0.000 | 0.390 | 0.010 |
| Gemma-2B + Prompt Tuning | 40,960 | 0.360 | 0.040 | 0.141 |
| Phi2-2.7B + Prompt Tuning | 51,200 | 0.000 | 0.690 | 0.000 |
| Mistral-7B v0.3 + Prompt Tuning | 81,920 | 0.360 | 0.360 | 0.434 |
| Gemma-2-2B + LoRA | 516,096 | 0.500 | 0.740 | 0.808 |
| Phi2-2.7B + LoRA | 70,272 | 0.520 | 0.710 | 0.778 |
| Mistral-7B v0.3 + LoRA | 73,728 | 0.180 | 0.690 | 0.798 |
| Gemma-2-2B + P Tuning | 97,208 | 0.590 | **0.800** | 0.808 |
| Phi2-2.7B + P Tuning | 129,464 | 0.460 | 0.720 | 0.808 |
| Mistral-7B v0.3 + P Tuning | 172,472 | 0.580 | 0.710 | 0.808 |
| Gemma-2-2B + **GraphToken** (MPNN) | 299,520 | **0.820** | **0.800** | **0.879** |

**Results.** The results of this experiment, summarized in Table 1, demonstrate that GraphToken significantly outperforms other options for encoding graph structure in PaLM-2-S on all graph, node, and edge-level tasks. Compared to PROMPT TUNING, we see that having access to the graph information provides a significant advantage for graph reasoning. Interestingly, after adding in GraphTokens, we can see that our small LLM (PaLM-2-S) is able to outperform its significantly larger sibling (PaLM-2-L) on 7 out of 9 tasks (the only task the large model seemed to retain a sizable advantage on was `connected nodes`.

### 4.2 EXPERIMENT 2: CHEMICAL PROPERTY PREDICTION

We next evaluate the performance of GraphToken on a suite of molecular property prediction tasks from the ChemLLMBench benchmarking suite (Guo et al., 2023b). A full description of both the dataset and the experimental setting is provided in §A.6.

**Baselines.** We evaluate GraphToken against a suite of other parameter-efficient finetuning methods, to isolate the performance improvement of GraphToken against other parameter-efficient methods by virtue of efficiently encoding structure into the input tokens for the LLM. We compare against LoRA (Hu et al., 2021), prompt tuning (Lester et al., 2021), and P tuning (Liu et al., 2023a), three established methods for parameter-efficient tuning.

**Results.** The results of this experiment, summarized in Table 3, demonstrate that GraphToken significantly outperforms existing parameter-efficient finetuning methods on molecular property prediction tasks. GraphToken outperforms the next-closest parameter-efficient baseline, P Tuning (Liu et al., 2023a), by up to 23% accuracy on molecular property prediction. Notably, GraphToken does well on the highly imbalanced ClinTox dataset, where 80.8% of samples belong to the majority class. Finally, we again see that GraphToken allows Gemma2-2B, to outperform a larger LLM (Mistral-7B) even when it is also augmented with other parameter-efficient finetuning methods.

## 4.3 ANALYSIS: GRAPH ENCODER DESIGN

From the results in Table 1, we can see that graph encoders can significantly improve a LLM's capability on graph reasoning tasks. However the choice of graph encoders has a significant effect on task performance. Here we study how different architecture choices affect the quality of the graph representation for a language model's use, including the choices of the graph convolution, the features available to the network, and the hyper-parameters.

### 4.3.1 CHOICE: GRAPH CONVOLUTION

This experiment investigates the impact of graph convolution choice on the performance of GraphToken. We evaluate the following diverse set of encoders:

- **Graph Convolutional Network (GCN)**: One of the earliest GNNs, this model does mean pooling of neighbor features, followed by a non-linearity (Kipf & Welling, 2017).
- **Message Passing Neural Network (MPNN)**: A generalization of the GCN, this model allows for more flexible aggregation of neighbor features, and has additional nonlinear feature transformations possible (Gilmer et al., 2017).
- **Graph Isomorphism Network (GIN)**: A GNN designed specifically to maximize the expressiveness of the model, w.r.t. a classic graph isomorphism test (Xu et al., 2018).
- **Multi-Head Attention (Graph Transformer)**: This GNN adapts transformer style attention, effectively learning a graph structure (Dwivedi & Bresson, 2021).
- **Heterogeneous Graph Transformer (HGT)**: Another adoption of transformer style attention (it can be applied to non-heterogeneous graphs as well) (Hu et al., 2020).
- **Linear Aggregation** In addition to the popular encoders from the literature, we also evaluated a family of models which use linear aggregation of features, as this has been shown to work surprisingly well on a number of tasks (Bojchevski et al., 2020).
  - **Node Set**: This model simply pools all the node features in the graph together.
  - **Edge Set**: This model simply pools all the edge features together (edge features are defined as the concatenation of its two nodes' features).

**Setting**: The experimental setup is similar to the experiment in §4.1.

**Result**: Table 2 shows the results for each model on GraphQA$_{\text{Test}}$. In general, we see that there is no one model that consistently dominates across graph encoding tasks. Instead, we see that different graph encoder architectures have strengths and weaknesses advantages.

There is one notable pattern however, the simple linear GNN models perform quite strongly at their respective counting tasks (i.e. NodeSet does well at `node count`, EdgeSet does well at `edge count`). However models with non-linear effects are still capable on these tasks (*e.g.*, MHA does well at `node count`, and MPNN does well on `edge count`).

### 4.3.2 CHOICE: GNN FEATURES

Recently, positional node encodings (Wang et al., 2022; Dwivedi et al., 2023; Lim et al., 2023) were proposed to enhance the expressivity of GNNs. On the other hand, in molecular modeling it has been shown recently that non-equivariant encoders can match or exceed quality of equivariant ones (Wang et al., 2023c). This raises a more general question: do GNNs need to be equivariant in order to generalize, especially with extremely powerful decoders, such as LLMs? We investigate this question by testing the graph reasoning capability of GraphToken with three distinct node featurization settings:

- **LPE**: Laplacian positional encodings using the normalized Laplacian matrix (Dwivedi et al., 2023).
- **IDX**: unique identity encoding designed to break the GNN equivariance.
- **LPE+IDX**: a concatenation of the above two strategies.

**Setting.** The experimental setup is similar to §4.3. Node features of dim $d = 4$ are evaluated for LPE and IDX featurization. Models using LPE+IDX contain node features of size $d = 8$.

**Result**. The outcome of this experiment are show in Figure 2, where we see the difference of all models from the SOFTPROMPT baseline (Lester et al., 2021) when evaluated on GraphQA$_{\text{Test}}$. The core result is that learned positional embeddings (Fig. 2b) generally outperform Laplacian position embeddings (Fig 2a) for most encoders and most tasks. These results show that breaking equivariance

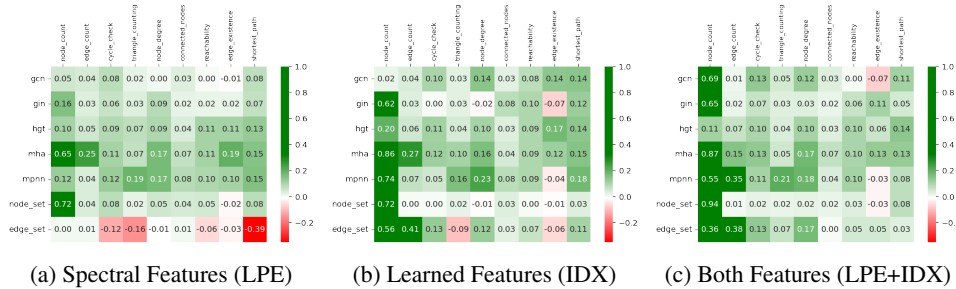

(a) Spectral Features (LPE)  (b) Learned Features (IDX)  (c) Both Features (LPE+IDX)

Figure 2: Effect of varying node features used in the graph encoder. Shown are performance delta from the PROMPT TUNING baseline on GraphQA$_{\text{Test}}$. We see that breaking equivariance via learned features (Fig. 2b) generally improve the model performance, but the combination of learned and spectral features (Fig. 2c) proves uniquely powerful for some encoders.

surprisingly adds additional capabilities for graph reasoning when powerful decoders are present. Some additional observations include:

- *Counting Tasks*. Learned features seem to provide essential lift for basic counting tasks (`node count`, `edge count`, and `node degree`) in many encoders.
- *Combination*. In some very interesting cases of task and encoder, the combination of both types of features greatly improved model performance (Fig. 2c). For example, GCN and NodeSet significantly improved at the `node count` task.
- *Linear models*. NodeSet (an encoder which does not use the graph edges) generally benefited from spectral features as they added previously unseen information about the graph structure.

### 4.3.3 PARAMETER USAGE IN GRAPHTOKEN FOR GRAPHQA

**Setting**: We consider the graph convolution evaluation from §4.3.1, using LPE features with dimensionality $d = 4$. The graph encoders have a latent space of size 128. We then project this into a prompt embedding with approximately $80,000$ parameters in GraphToken .

**Results**:

Table 4 shows the number of parameters used in the graph encoder. Here 'Body' refers to the number of parameters in the graph encoder itself, and 'Head' refers to the parameters in the transformation layer to the higher-dimensional LLM token space.

Its also insightful to consider the number of parameters used in each of the models. Table 4 specifies total number of parameters used by each GNN architecture. We note that this size is dominated by the total number of parameters in the projection layer to the token space (roughly

Table 4: # of parameters in the graph encoder.

| Encoder | Body | Head |
|---------|------|------|
| GCN | 17,152 | $1.1 \times 10^7$ |
| GIN | 17,152 | $1.1 \times 10^7$ |
| MPNN | 83,968 | $1.1 \times 10^7$ |
| HGT | 198,788 | $1.1 \times 10^7$ |
| MHA | 101,376 | $1.1 \times 10^7$ |
| Node Set | 0 | $4.1 \times 10^5$ |
| Edge Set | 0 | $7.4 \times 10^5$ |

11 million). Out of all non-linear architectures, attention-based ones (MHA and HGT) add the most encoder-based parameters. In general, the size of our graph encoder models varies from 17k to 199k parameters. This is *significantly smaller* than typical LLMs, which currently often contain tens or hundreds of *billions* of parameters. For example, the open-source LLama2 language model scales from 7 billion to 70 billion parameters (Touvron et al., 2023). Meanwhile the closed source PaLM 1 model contains 540 billion parameters (Chowdhery et al., 2022). In light of this, we can see that GraphToken is highly parameter-efficient, and significantly improves the graph reasoning capability of a LLM while barely adding any parameters at all.

Table 5: Predicting **bipartiteness** using graph encoders trained for different tasks, as measured by AUC×100 on all graphs with 8 nodes. Observe that graph encoders trained on `cycle check` and `triangle count` generalize well to bipartiteness detection.

| | Method | Node count | Edge count | Cycle check | Triangle count | Node degree | Connected nodes | Reachability | Edge existence | Shortest path |
|---|---|---|---|---|---|---|---|---|---|---|
| | | | | | | **Original GraphToken Encoder Training Task:** | | | | |
| Non-linear | GCN | 53.82 | 53.28 | 55.46 | 50.00 | 50.00 | 54.64 | 50.00 | 48.48 | 51.60 |
| | GIN | 51.09 | 53.27 | 52.74 | 51.91 | 53.26 | 53.57 | 51.36 | 52.17 | 52.18 |
| | MPNN | **68.01** | **71.34** | 56.82 | 76.82 | **60.13** | 60.95 | 61.77 | 62.58 | 54.37 |
| | HGT | 50.00 | 54.35 | 68.53 | **95.03** | 50.27 | 59.81 | **68.85** | **74.58** | 50.00 |
| | MHA | 50.27 | 66.39 | **87.00** | 72.14 | 58.74 | **66.38** | 51.63 | 54.12 | **64.45** |
| Linear | Node Set | 56.55 | 57.38 | 56.30 | 55.74 | 56.29 | 56.28 | 55.73 | 57.93 | 56.56 |
| | Edge Set | 50.82 | 50.82 | 50.82 | 50.55 | 50.54 | 50.54 | 50.82 | 50.82 | 50.54 |

## 5 DISCUSSION

So far we have examined the performance benefits of GraphToken, and the design choices necessary when building a graph encoder. However several questions remain: (1) What exactly are the encoders doing, and (2) does it generalize? In this section we seek to provide some insight (if not answers) to these questions, and lay the foundations for future work.

### 5.1 GENERALIZATION OF GRAPH ENCODERS

This section studies the properties learned by GraphToken's graph encoders by directly examining the representations they produce. We study both the in-distribution and out-of-distribution properties of these encoders. We briefly discuss one example here and present additional results in §A.9.

**Setting:** For the generalization experiment, we consider 9 additional tasks: total number of edges, maximum node degree, graph diameter, number of triangles, average local clustering coefficient, largest core number, average shortest path length, testing planarity, and testing bipartiteness.

The evaluation goes as follows: First, we train an encoder on a task from GraphQA (*e.g.*, `cycle check`). Then, to evaluate the cross-task generalizability of the different encoders we train a kNN classifier (or regressor) with $k = 5$ on the representations of (i) an exhaustive set of connected graphs with 8 nodes (called `graph8c` in Balcilar et al. (2021)) and (ii) an exhaustive set of tree graphs with 15 nodes. We note that because we are generating a large set of graphs (*e.g.*, there are 11117 graphs of size 8) and only trained on GraphQA$_{\text{Train}}$ (1000 instances), the vast majority of the graphs we are using here are unseen. As an illustration, a UMAP (McInnes et al., 2018) visualization of the embeddings for all 8 node graphs using two GNN encoders is presented in Figure 6.

The graphs are generated by enumerating all graphs of a given size exhaustively. We use `geng` (McKay et al., 1981) to generate these graphs.

**Results**. We focus here on the task of predicting whether a graph is bipartite. From basic graph theory we know that if there is a triangle or an odd cycle in a graph it can not be bipartite. Therefore, we expect `triangle count` and `cycle check` training objectives to perform well on this task. In Table 5 we can see that this is precisely what happens, with attention-based methods taking the lead. This is an interesting example of *generalization* from the graph encoders to a new task.

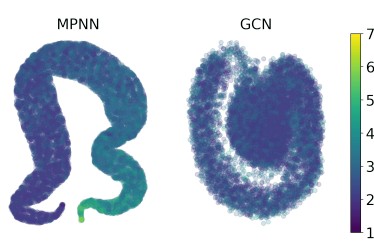

Figure 3: UMAP (McInnes et al., 2018) projection of GraphToken embeddings produced by two different encoders, colored by the diameter of a graph. We plot all 8-node graphs.

Overall, there is a significant performance gap between different graph encoders. We observe significant correlations in performance of in-distribution learning – for instance, GraphToken trained on `edge count` performs the best on `edge count` prediction. What is interesting is that `node count` performs comparably here. This suggests that graph encoders learn some universal features that are applicable to many different downstream tasks.

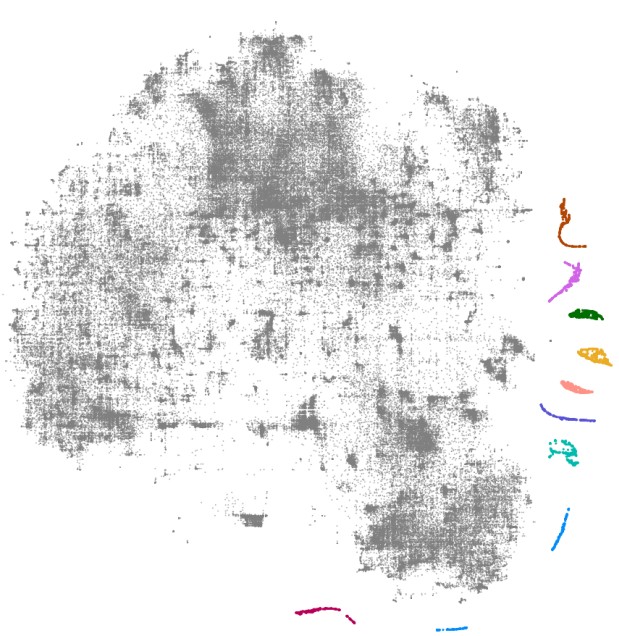

Figure 4: t-SNE visualization of the Gemma2 token embeddings (grey), and learned GraphTokens for 9 GraphQA tasks: ● connected nodes; ● cycle check; ● edge count; ● edge existence; ● node count; ● node degree; ● reachability; ● shortest path; ● triangle counting.

### 5.2 VISUALIZATION OF GRAPH ENCODINGS

In order to better understand the latent representations giving GraphToken such strong performance on graph reasoning tasks, we collected the activations for all 9 GraphQA tasks using a MPNN GraphToken encoder. They are visualized using t-SNE (Van der Maaten & Hinton, 2008) in Figure 4. We observe that each task learns a specific set of soft tokens that are more similar within than across tasks. Furthermore, when projected down to the space of language tokens, they exist in an underutilized region of the space – the nearest neighbor's cosine distance is 0.7. This sheds additional light on how the model adapts for an unfamiliar task.

## 6 CONCLUSIONS

In this work we have studied the structured data encoding problem for LLMs. Our novel method, GraphToken, learns a graph embedding function through the gradients provided by a *frozen* LLM. GraphToken is especially well suited for projecting structured data into latent 'prompt space'. It is a parameter-efficient method as it requires only training the graph encoder and does not update LLM parameters. Our extensive experimental analysis across 9 graph reasoning tasks shows that GraphToken greatly improves graph reasoning in LLMs – we observe up to a 73% improvement on the GraphQA benchmark. We also illustrated how GraphToken offered significant improvements to the Gemma2-2B model on molecular property prediction tasks, presenting strong performance against other PEFT methods, and even exceeding the performance of a larger 7B parameter model.

There is still much to do! We believe that our approach is fundamental for adapting new structured data sources to LLMs (which are expensive and time consuming to train), and presents an attractive way of improving fundamental problems in LLMs, including *hallucinations*, *factuality*, and *freshness*.

# 7 REPRODUCIBILITY STATEMENT

**Note:** Our model GraphToken requires gradient access to a LLM in order to train. The primary experiments in this paper are conducted on PaLM 2, a large model with proprietary weights. We believe that understanding how structure can best be incorporated into large models is an important area of work, and that our results show how our parameter efficient method can drastically increase the graph reasoning capabilities of a truly large language model.

However, we also believe that it is important for research to be accessible. To that end, we have developed a reference implementation compatible with smaller, open-weight models (i.e. *Gemma2-2B*). Experiments using this implementation for molecular property prediction are shown in Section 4.2. We will release this implementation and tutorial materials designed to facilitate the training and use of this reference implementation with smaller LMs upon the paper's acceptance.

**Source code:** To accelerate future research, we will open-source a reference version of GraphToken upon acceptance of the paper.

**Graph Encoders** The PaLM 2 GraphToken we describe was implemented in Tensorflow (Abadi et al., 2015) using the TF-GNN library (Ferludin et al., 2023). Many of the graph encoders studied in this paper are reference implementations of graph convolutions available in this TensorFlow library at `https://github.com/tensorflow/gnn/tree/main/tensorflow_gnn/models`.

**Large Language Model:** The largest LLM used in our experiments is the instruction-fine-tuned Flan (Chung et al., 2022) checkpoint of PaLM 2 S (Anil et al., 2023). A PaLM 2 API is available through VertexAI at `https://python.langchain.com/docs/integrations/llms/google_vertex_ai_palm.html`. For the smaller open weight models used in our experiments (Gemma2, Phi2, and Mistral-7b), we used weights and code from HuggingFace (Wolf, 2019).

**Accelerator usage:** Experiments were carried out on Google TPUv3 and TPUv5e (Jouppi et al., 2017).

**GraphToken Training:** Our training procedure is very similar to that used by soft prompting methods (Lester et al., 2021). The training input consists of triples $(G, T, A)$ where $G$ is a graph structure, $T$ is a statement or question describing the task (*e.g.*, 'Does this graph contain a cycle?' for `cycle check`) and $A$ is the ground truth answer ('Yes, there exists a cycle in this graph.').

In the forward pass, we compute the augmented query $Q = \mathcal{E}(G) || \mathcal{T}(T)$, concatenating the Graph-Token encoding of the graph $\mathcal{E}(G)$ with the initial embedding of the task textual representation, $\mathcal{T}(T)$.

We train by optimizing the final LLM perplexity (total log-likelihood), $\mathcal{L}(A \mid Q)$, of the expected answer $A$ with respect to the augmented query, $Q$. We minimize this loss, back-propagating the gradient of $\mathcal{L}$ with respect to $\mathcal{E}(G)$ to the parameters of the GraphToken encoder – keeping all LLM parameters frozen. We use the Lion optimizer (Chen et al., 2023a) with a learning rate $\alpha = 0.05$.

**Model Selection:** In order to select the best hyper-parameters for the graph encoder, we used the loss on the training dataset (GraphQA$_{\text{Train}}$) to perform model selection. Unless specified otherwise, we report the corresponding test scores (from GraphQA$_{\text{Test}}$).

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

## A  APPENDIX

### A.1  ADDITIONAL RELATED WORK

We briefly discuss several additional areas of tangentially related work for completeness.

**'GNN-only' Pretraining** While there is a growing body of work in pre-training, fine-tuning, and prompt-tuning with GNNs by themselves  (Fang et al., 2023; Liu et al., 2023b), the research, though conceptually similar, differs crucially from our work.  GNN-based approaches lack the textual understanding capabilities that are central to the integration of LLMs with graph learning and reasoning.

**Soft Prompt Functions for Images** The idea of parameterizable prompts has been explored in the visual domain Tsimpoukelli et al. (2021); Alayrac et al. (2022); Luo et al. (2022); Chen et al. (2022a); Li et al. (2023); Moon et al. (2023). In this light, our work could be viewed as an extension of these ideas to the modality of graph structured ideas. We note that the visual domain has been studied substantially more than graph neural networks (which are a younger field), and that developing a soft prompt function for graph structured data introduces a number of additional challenges that we describe in this work.

**Abstract Parse Information** Additionally some work has studied adding semantic parse trees in order to semantically improve generation (Mager et al., 2020; Ribeiro et al., 2021). Unlike these works which focus on AMR graphs, we focus on more general graph reasoning capabilities and downstream tasks (such as molecular property prediction) that depend on graph structure. Understanding relational structure is key for these tasks.

**Concurrent Work** Due to the importance of this research area, there have recently been a number of concurrent works seeking to combine graph structure and large language models. GraphWiz (Chen et al., 2024a) proposes an instruction following LM for graph reasoning, showing strong improvements over SFT tuning of regular language models for textual reasoning over graphs. Similarly, GraphGPT (Tang et al., 2024) uses an instruction tuning recipe, along with a frozen graph tokenizer to improve classic graph learning tasks (node classification, link prediction, etc). Additionally, LLaGA (Kong et al., 2024) and UniGraph (Chen et al., 2024b) propose different ways of aligning graph structure and natural language.

Despite the growing amount of related work, we note that GraphToken still has significant advantages. Much of the concurrent work focuses on tuning entire language models – our work shows that this is frequently unnecessary, and competitive results can be achieved via our parameter efficient method. In addition, we were the first (to the best of our knowledge) to propose soft prompting via GNN encoders learned from LLM gradients.

### A.2  LIMITATIONS AND FUTURE WORK

#### A.2.1  LIMITATIONS

Our results have shown that GraphToken is both a flexible and generalizable encoder of graph structured data for LLMs. Here we discuss some limitations of the method as inspiration for future work.

**Encoder Generalizability**.  The main limitation of GraphToken is that its encoder might learn spurious correlations due to idiosyncrasies in the distribution of input graphs it was trained on. As such, it's important that the encoder works robustly regardless if its evaluated on a different distribution of input graphs (w.r.t. their density, number of nodes/edges, etc). We note that this is a general weakness of GNNs and not specific to GraphToken itself. As such, there is a rich literature on creating robust GNNs (Zhang et al., 2024) that has made significant progress in creating more generalizable GNN architectures. We expect that these results will be directly able to be "plugged in" to GraphToken encoders and will greatly aid in their generalization.

#### A.2.2  FUTURE WORK

This work opens up an exciting new avenue of exploration for reasoning with structured data and LLMs. Some potential avenues that we consider particularly exciting include:

- This work considers existing convolutions and measures their effectiveness. An obvious and essential next step is designing graph convolutions that best support LLMs in various graph reasoning tasks.
- Evaluating the usefulness of this approach for factual grounding. Can we improve the ability of an LLM to answer questions about the data using prompting over knowledge graphs? Could an LLM answer novel questions about a molecule given a GNN-produced representation of it?
- GraphToken improves performance with broken equivariance. Can this result inform other problems with very strong decoder models?
- This work examines how a GNN can be used to an enhance LLMs, but what about the reverse? Can we use an LLM to interrogate a GNN to better explain its results or provide higher quality answers?

### A.3 BASELINES

To rigorously evaluate the performance of GraphToken, we compare it against the following established baselines for prompt optimization:

- ZERO-SHOT. In this approach, the model is given a task description and immediately asked to produce the desired output. No additional examples or demonstrations are provided.
- FEW-SHOT. This approach provides the model with a few examples of the task and their desired outputs (Brown et al., 2020). Unlike traditional training, these examples are included directly in the prompt, allowing the model to learn and adapt during the inference.
- CoT. Chain-of-thought (CoT) prompting (Wei et al., 2022) provides examples each showing step-by-step reasoning, teaching the LLM to generate its own thought processes for tackling new tasks.
- ZERO-COT. Zero-shot CoT (Kojima et al., 2022) builds upon Chain-of-Thought (CoT) prompting by eliminating the need for training examples. The LLM generates its own step-by-step reasoning process using a simple trigger phrase like "Let's think step by step".
- COT-BAG. BAG prompting (Wang et al., 2023b) extends COT to improve the performance of LLMs on graph-related tasks by appending "Let's construct a graph with the nodes and edges first" to the prompt.
- SOFT-PROMPT. This approach uses the standard soft prompt tuning of Lester et al. (2021). It optimizes a global *static* prompt which is shared across problem instances to improve task performance. Unlike our proposed method, it does not have access to the graph information, making the results of this approach equivalent to that of a majority classifier.

### A.4 GRAPH ENCODERS

**Notation.** We briefly describe the notation we will use. The graph $G = (V, E)$ contains the set of $V$ nodes and $E$ edges. While we will only discuss simple graphs, everything discussed can be extended to heterogeneous graphs w.l.o.g. (Battaglia et al., 2018; Ferludin et al., 2023).

Using the notation of Ferludin et al. (2023), a GNN has two primary operations. First, a next state function (NEXTSTATE) which computes the hidden state $\mathbf{h}_v$ of a node (or edge, $\mathbf{m}_{(u,v)}$) given information from its neighbors and its previous state, and an aggregation function (EDGEPOOL) which pools information for a node's immediate neighborhood into a fixed size representation. More formally, we can say that the next state of a node is:

$$\mathbf{h}_v^{(i+1)} = \text{NEXTSTATE}_V^{(i+1)}(\mathbf{h}_v^{(i)}, \overline{\mathbf{m}}_v^{(i+1)}).$$

Then the pooled messages $\overline{\mathbf{m}}_v^{(i+1)}$ are defined as follows:

$$\mathbf{m}_{(u,v)}^{(i+1)} = \text{NEXTSTATE}_E^{(i+1)}(\mathbf{h}_u^{(i)}, \mathbf{h}_v^{(i)}, \mathbf{m}_{(u,v)}^{(i)}),$$

$$\overline{\mathbf{m}}_v^{(i+1)} = \text{EDGEPOOL}^{(i+1)}(\mathbf{h}_v^{(i)}, \{\mathbf{m}_{(u,v)}^{(i+1)} \mid u \in \mathcal{N}(v)\}).$$

Different realizations of the NEXTSTATE and EDGEPOOL functions can implement a wide variety of GNN operations. This can include powerful models which use Transformer style attention instead of the provided graph edges (Dwivedi & Bresson, 2021).

The architecture of NodeSet and EdgeSet is shown in Figure 5. Other GNN models have graph convolutions before node/edge states are read out.

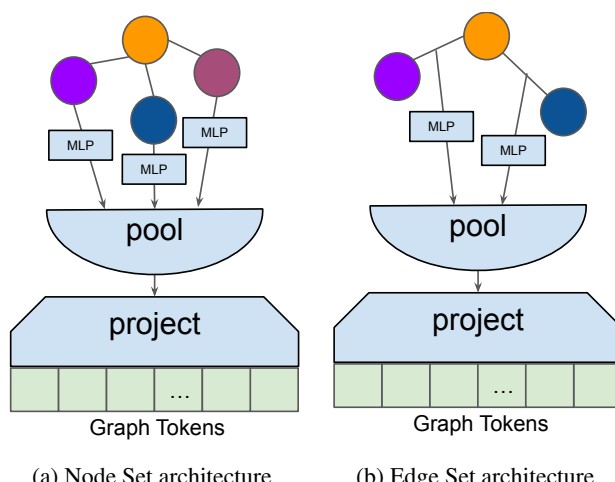

(a) Node Set architecture          (b) Edge Set architecture

Figure 5: Figurative illustrations of set-based GNN architectures employed in the paper. We pool representations from either nodes or edges, transform them via an MLP with shared weights, pool, and project to the GraphToken space.

## A.5 DATASET STATISTICS

The graphs used in the experiments in this paper and the corresponding graph reasoning tasks are taken from Fatemi et al. (2024). There are $1,000$ graphs in the train set and $500$ graphs in the test set. The graphs are generated randomly using Erdős-Rényi (ER) random graph model (Erdős & Rényi, 1959). Graph size ranges from 5 to 20 nodes.

**Train set statistics.**

- **Average number of nodes:** $11.90$
- **Average number of edges:** $37.01$
- **Average node degree:** $5.43$

**Test set statistics.**

- **Average number of nodes:** $12.37$
- **Average number of edges:** $39.79$
- **Average node degree:** $5.70$

To address dataset size limitations in the benchmark, we evaluate model generalization abilities on an exhaustive collection of connected 8-node graphs. This allows us to test within-data-distribution generalization to both unseen data and unseen tasks (*e.g.*, bipartiteness in Table 5. To test out-of-distribution generalization, we verify GraphToken works well on tree-structure graphs, again, generated exhaustively. Note that for tree graphs, the number of edges is the lowest possible for a connected graph, leading to more generalization challenges.

## A.6 MOLECULAR PROPERTY PREDICTION WITH GRAPHTOKEN

Here we elaborate on the experimental settings from Section 4.2.

**Datasets.** The benchmarking datasets within ChemLLMBench (Guo et al., 2023b) are a collection of datasets intended to test prediction of different molecular properties, with many of the property prediction datasets originating from MoleculeNet (Wu et al., 2018). The Blood-Brain Barrier Penetration (BBBP) dataset tests the ability of compounds to penetrate the blood-brain barrier. The BACE dataset tests inhibition of human $\beta$-secretase 1, while the Tox21 dataset tests the toxicity of compounds on 12 different nuclear receptor and stress receptor targets. The ClinTox dataset compares FDA-approved drugs to failed drugs for toxicity. For all property prediction tasks, we evaluate the accuracy of model predictions against ground truth molecular property labels.

**Processing.** In this section, we overview the encoding method used for chemical molecules in the ChemLLMBench (Guo et al., 2023b) datasets. At a high-level, a molecule is taken as input and converted into a graph where nodes represent atoms connected by edges that represent atomic bonds. The GNN encoder component encodes the graph and outputs tokens which are prepended to the LLM input prompt, which the LLM can then reason over in chemistry tasks.

To encode structural information about chemical molecules, we follow a principled framework for encoding node, edge, and graph-level features pertaining to input molecular graphs:

1. *Node-level features* comprise of atomic properties of atoms which are represented by nodes in the input molecular graph. These node feature capture properties such as atomic number, mass, and charge.
2. *Edge-level features* comprise of information about bonds which connect atoms together in the molecular graph. Edge features give information about the type of bond linking atoms together.
3. *Graph-level features* consist of molecular signatures, or fingerprints, of the molecule. These features are obtained through standardized molecular fingerprint generators such as mordred (Moriwaki et al., 2018), which have predefined molecular descriptors which capture properties about input molecules such as the presence or count of certain subgroups.

Given this input feature space, we utilize a GNN encoder to perform message-passing over the molecular graph, and perform a pooling readout operation across node tokens and the graph-level molecular fingerprint features in order to produce the final readout tokens.

**Baselines.** We evaluate GraphToken against a suite of other parameter-efficient finetuning methods, to isolate the performance improvement of GraphToken against other parameter-efficient methods by virtue of efficiently encoding structure into the input tokens for the LLM. We compare against LoRA (Hu et al., 2021), prompt tuning (Lester et al., 2021), and P tuning (Liu et al., 2023a), three established methods for parameter-efficient tuning.

**Results.** The results of this experiment, summarized in Table 3, demonstrate that GraphToken significantly outperforms existing parameter-efficient finetuning methods on molecular property prediction tasks. GraphToken outperforms the next-closest parameter-efficient baseline, P Tuning (Liu et al., 2023a), by up to 23% accuracy on molecular property prediction. Notably, GraphToken does well on the highly imbalanced ClinTox dataset, where 80.8% of samples belong to the majority class. Finally, we again see that GraphToken allows Gemma2-2B, to outperform a larger LLM (Mistral-7B) even when it is also augmented with other parameter-efficient finetuning methods.

A.6.1 RUNNING TIME

It is instructive to also consider the running time of these various PeFT methods on the molecular property prediction task. The timing of these results is available below:

Table 6: Molecular property prediction running times (**measured by hours**). GraphToken with Gemma2-2B is the fastest method (in addition to the highest performing).

| | Trainable parameters | BACE | BBBP | ClinTox |
|---|---|---|---|---|
| Gemma-2B + Prompt Tuning | 40,960 | 1.55 | 2.14 | 1.49 |
| Phi2-2.7B + Prompt Tuning | 51,200 | 1.63 | 2.27 | 1.71 |
| Mistral-7B v0.3 + Prompt Tuning | 81,920 | 3.98 | 3.97 | 3.99 |
| Gemma-2-2B + LoRA | 516,096 | 0.30 | 0.47 | 0.28 |
| Phi2-2.7B + LoRA | 70,272 | 0.37 | 0.98 | 0.57 |
| Mistral-7B v0.3 + LoRA | 73,728 | 0.77 | 1.07 | 0.76 |
| Gemma-2-2B + P Tuning | 97,208 | 13.92 | 13.90 | 13.92 |
| Phi2-2.7B + P Tuning | 129,464 | 22.37 | 18.03 | 15.42 |
| Mistral-7B v0.3 + P Tuning | 172,472 | 23.95 | 23.89 | 23.87 |
| Gemma-2-2B + **GraphToken** (MPNN) | 299,520 | **0.13** | **0.27** | **0.19** |

These results show that GraphToken is not only the most performant model for the task, but also the fastest.

## A.7 MULTI-TASK GRAPH ENCODERS

It is also interesting to study whether the representations learned by GraphToken can generalize to new graph reasoning tasks at the LLM-level (beyond the embedding space investigation of §5.1).

In order to study this, we designed the following experiment. First, we trained a Multi-Task Graph-Token model on 9 of the 10 GraphQA tasks using Gemma2-2B as the LLM backbone. The same encoder is used for all tasks, and the graph encoder has no knowledge about the task while encoding. We then evaluate this model on a withheld task – *cycle check*. It has never seen cycle check before this eval.

**Results**: For context, we provide the single task performance ("SingleTask GraphToken") on Gemma2-2B as well as its text-only performance

|             | MultiTask GraphToken | SingleTask GraphToken | Gemma2-2B (text only) |
|-------------|:--------------------:|:---------------------:|:---------------------:|
| cycle check | 88.4                 | 98.8                  | 60.0                  |

Table 7: Graph Reasoning Generalization Experiment

We see that while there is a drop on performance (as might be expected) compared to optimizing a task directly: (1) GraphToken can generalize to a unseen task, and the generalization outperforms large models (e.g. PaLM-2-L with 83.3) on the task. (2) The generalization is much better than using a text representation of the graph with the backbone LLM.

## A.8 DOES GRAPH STRUCTURE MATTER?

It is interesting to study how graph structure might affect a GraphToken encoder's performance on downstream tasks. Are more complicated graphs harder? Do other structural patterns influence its results?

**Experiment design**. We use the Gemma2-2B multi-task encoder from §A.7 which is trained on 9 out of 10 GraphQA tasks. Then we calculated a number of graph properties that have been shown useful for analyzing GNN performance (Palowitch et al., 2022), and examined the correlation between these graph characteristics and whether the model was able to correctly answer its task (out of 10 GraphQA tasks) on unseen graphs. The results are as follows:

| Graph Property | Pearson correlation |
|----------------|:-------------------:|
| number nodes | -0.146 |
| number edges | -0.0581 |
| edge density | 0.0732 |
| degree gini | -0.104 |
| average degree | -0.040 |
| average clustering coefficient | 0.0111 |
| transitivity | 0.0103 |
| number of triangles | 0.00012 |
| connected component sizes | -0.0344 |

Table 8: The correlation of structure with GraphToken's performance on GraphQA tasks.

Interestingly, we find that most graph properties are uncorrelated with the downstream task's performance. We do see a weak negative correlation between the number of nodes in the graph and correctly answering tasks. However this is expected – as the number of nodes grows, the graph has the potential for more complexity.

These results support the strong generalization capabilities of GraphToken.

## A.9 Graph Encoder Generalization

### A.9.1 Experiment Design

**Setting:** For the generalization experiment, we consider 9 tasks in total: total number of edges; maximum node degree; graph diameter; number of triangles; average local clustering coefficient; largest core number; average shortest path length; testing planarity; testing bipartiteness.

The evaluation goes as follows: First, we train an encoder on a task from GraphQA (*e.g.*, `cycle check`). Then, to evaluate the cross-task generalizability of the different encoders we train a kNN classifier (or regressor) with $k = 5$ on the representations of (i) an exhaustive set of connected graphs with 8 nodes (called `graph8c` in Balcilar et al. (2021)) and (ii) an exhaustive set of tree graphs with 15 nodes. We note that because we are generating a large set of graphs (*e.g.*, there are 11117 graphs of size 8) and only trained on $\text{GraphQA}_{\text{Train}}$ (1000 instances), the vast majority of the graphs we are using here are unseen. As an illustration, a UMAP (McInnes et al., 2018) visualization of the embeddings for all 8 node graphs using two GNN encoders is presented in Figure 6.

The graphs are generated by enumerating all graphs of a given size exhaustively. We use `geng` (McKay et al., 1981) to generate these graphs.

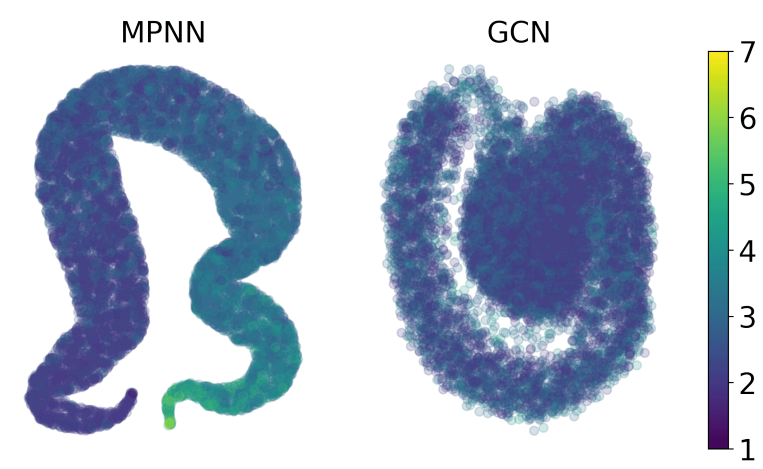

Figure 6: UMAP (McInnes et al., 2018) projection of GraphToken embeddings produced by two different encoders, colored by the diameter of a graph. We plot all 8-node graphs.

### A.9.2 Additional Results

We present additional results for graph encoder analysis. Tables 9–19 present additional results on more graph properties, as well as experiments on tree-structured graphs of size 15. In general, complete graph populations demonstrate significantly better performance than trees – we can attribute that to the fact that GraphToken was trained on diverse sets of data, and trees are somewhat out-of-distribution. Nevertheless, for all considered cases the best overall encoder model achieved better results than naïve set encodings.

Table 9: Average local clustering coefficient MSE measured on all connected graphs with 8 nodes. We highlight the best performance per training task in columns.

| | | Original GraphToken Encoder Training Task: | | | | | | | | |
|---|---|---|---|---|---|---|---|---|---|---|
| | **Method** | **Node count** | **Edge count** | **Cycle check** | **Triangle counting** | **Node degree** | **Connected nodes** | **Reachability** | **Edge existence** | **Shortest path** |
| Non-linear | GCN | 1.62 | 1.67 | 2.12 | 4.49 | 4.49 | 1.73 | 4.49 | 16.57 | 3.75 |
| | GIN | 2.18 | 2.29 | 2.45 | 2.60 | 2.44 | 2.31 | 3.73 | 2.88 | 3.37 |
| | MPNN | **1.03** | **0.95** | 1.38 | **0.81** | **1.50** | 1.34 | **1.68** | 1.87 | 1.47 |
| | HGT | 2.63 | 2.25 | 2.08 | 1.23 | 2.49 | 2.17 | 1.90 | 1.62 | 2.52 |
| | MHA | 2.69 | 1.01 | **1.23** | 0.96 | 1.56 | **1.25** | 2.08 | **1.59** | **1.29** |
| Linear | Node Set | 2.59 | 2.56 | 2.59 | 2.59 | 2.58 | 2.60 | 2.58 | 2.58 | 2.56 |
| | Edge Set | 2.22 | 2.22 | 2.22 | 2.22 | 2.24 | 2.23 | 2.22 | 2.22 | 2.23 |

Table 10: Degree accuracy on all connected graphs with 8 nodes. We highlight the best performance per training task in columns.

| | | Original GraphToken Encoder Training Task: | | | | | | | | |
|---|---|---|---|---|---|---|---|---|---|---|
| | **Method** | **Node count** | **Edge count** | **Cycle check** | **Triangle counting** | **Node degree** | **Connected nodes** | **Reachability** | **Edge existence** | **Shortest path** |
| Non-linear | GCN | 57.46 | 56.65 | 52.46 | 40.09 | 40.09 | 57.42 | 40.09 | 15.73 | 40.26 |
| | GIN | 56.86 | 56.30 | 54.55 | 48.75 | 55.59 | 57.56 | 40.14 | 50.81 | 44.83 |
| | MPNN | **69.45** | **69.60** | **67.19** | **71.84** | **64.56** | **67.62** | 61.37 | 58.66 | 63.18 |
| | HGT | 55.20 | 55.70 | 56.54 | 60.17 | 56.62 | 57.65 | 58.02 | 59.06 | 55.46 |
| | MHA | 54.86 | 64.33 | 62.86 | 65.63 | 61.67 | 63.22 | 56.98 | 61.60 | **63.97** |
| Linear | Node Set | 54.66 | 54.91 | 54.98 | 55.06 | 54.78 | 54.64 | 54.50 | 54.94 | 54.72 |
| | Edge Set | 63.48 | 63.37 | 63.07 | 63.55 | 63.08 | 63.37 | **63.47** | **63.06** | 63.44 |

Table 11: Diameter Accuracy on all connected graphs with 8 nodes. We highlight the best performance per training task in columns.

| | | Original GraphToken Encoder Training Task: | | | | | | | | |
|---|---|---|---|---|---|---|---|---|---|---|
| | **Method** | **Node count** | **Edge count** | **Cycle check** | **Triangle counting** | **Node degree** | **Connected nodes** | **Reachability** | **Edge existence** | **Shortest path** |
| Non-linear | GCN | 66.86 | 67.81 | 66.70 | 37.37 | 37.37 | 68.91 | 37.37 | 52.13 | 55.13 |
| | GIN | 66.06 | 64.87 | 63.97 | 61.09 | 64.98 | 66.43 | 37.80 | 60.65 | 54.82 |
| | MPNN | **76.92** | **76.86** | 73.63 | **78.33** | **74.78** | **77.18** | **74.42** | **69.56** | **76.23** |
| | HGT | 63.97 | 65.24 | 66.88 | 70.45 | 65.30 | 68.45 | 69.64 | 68.97 | 66.04 |
| | MHA | 63.76 | 74.17 | **76.00** | 74.03 | 73.50 | 74.71 | 68.45 | 69.32 | 72.95 |
| Linear | Node Set | 67.28 | 67.24 | 67.01 | 66.97 | 66.81 | 67.19 | 67.09 | 66.87 | 66.79 |
| | Edge Set | 66.99 | 66.51 | 66.63 | 66.83 | 66.65 | 67.02 | 66.60 | 66.93 | 66.90 |

Table 12: k-Core Accuracy on all connected graphs with 8 nodes. We highlight the best performance per training task in columns.

| | | Original GraphToken Encoder Training Task: | | | | | | | | |
|---|---|---|---|---|---|---|---|---|---|---|
| | **Method** | **Node count** | **Edge count** | **Cycle check** | **Triangle counting** | **Node degree** | **Connected nodes** | **Reachability** | **Edge existence** | **Shortest path** |
| Non-linear | GCN | 69.49 | 69.15 | 66.61 | 58.33 | 58.33 | 69.16 | 58.33 | 25.18 | 61.55 |
| | GIN | 68.03 | 65.98 | 64.85 | 62.67 | 66.74 | 67.84 | 58.84 | 63.34 | 59.08 |
| | MPNN | **87.42** | **87.54** | **81.81** | **88.63** | **80.30** | **83.48** | **80.08** | 71.01 | **82.05** |
| | HGT | 63.92 | 65.29 | 67.00 | 70.01 | 65.44 | 67.32 | 68.35 | 70.08 | 65.13 |
| | MHA | 64.30 | 80.80 | 73.49 | 80.81 | 76.98 | 78.83 | 69.43 | **74.21** | 75.92 |
| Linear | Node Set | 68.23 | 68.74 | 68.50 | 68.71 | 68.07 | 67.99 | 68.85 | 68.17 | 68.70 |
| | Edge Set | 66.30 | 65.78 | 65.58 | 66.15 | 65.76 | 65.91 | 65.94 | 65.77 | 65.71 |

Table 13: #edges Accuracy on all connected graphs with 8 nodes. We highlight the best performance per training task in columns.

| | | Original GraphToken Encoder Training Task: | | | | | | | | |
|---|---|---|---|---|---|---|---|---|---|---|
| | **Method** | **Node count** | **Edge count** | **Cycle check** | **Triangle counting** | **Node degree** | **Connected nodes** | **Reachability** | **Edge existence** | **Shortest path** |
| Non-linear | GCN | 38.91 | 39.19 | 35.94 | 11.60 | 11.60 | 40.24 | 11.60 | 2.19 | 14.58 |
| | GIN | 38.13 | 37.33 | 36.57 | 31.66 | 37.74 | 38.34 | 11.88 | 31.45 | 25.92 |
| | MPNN | **86.58** | **86.72** | **53.15** | **84.56** | **52.12** | **66.01** | **50.70** | **41.96** | **59.95** |
| | HGT | 35.63 | 37.45 | 38.23 | 40.39 | 37.14 | 37.80 | 39.68 | 39.74 | 36.86 |
| | MHA | 35.85 | 55.32 | 45.04 | 53.52 | 47.89 | 49.44 | 39.69 | 42.84 | 46.17 |
| Linear | Node Set | 40.06 | 40.14 | 39.40 | 40.15 | 39.97 | 39.72 | 39.88 | 39.79 | 39.89 |
| | Edge Set | 37.93 | 38.11 | 38.05 | 37.92 | 38.05 | 37.67 | 37.64 | 37.82 | 37.91 |

Table 14: Planarity AUC on all connected graphs with 8 nodes. We highlight the best performance per training task in columns.

| | | Original GraphToken Encoder Training Task: | | | | | | | | |
|---|---|---|---|---|---|---|---|---|---|---|
| | **Method** | **Node count** | **Edge count** | **Cycle check** | **Triangle counting** | **Node degree** | **Connected nodes** | **Reachability** | **Edge existence** | **Shortest path** |
| Non-linear | GCN | 74.18 | 73.76 | 72.61 | 50.00 | 50.00 | 74.74 | 50.00 | 50.00 | 49.44 |
| | GIN | 77.35 | 73.00 | 72.06 | 69.37 | 74.86 | 75.85 | 50.73 | 68.97 | 61.58 |
| | MPNN | **86.14** | **86.52** | **84.16** | **86.64** | **83.74** | **85.17** | **84.32** | 77.84 | **85.55** |
| | HGT | 69.24 | 71.41 | 71.02 | 74.07 | 71.47 | 72.20 | 72.20 | 73.59 | 71.55 |
| | MHA | 69.96 | 80.87 | 78.35 | 80.46 | 81.53 | 81.21 | 74.98 | **78.29** | 80.58 |
| Linear | Node Set | 78.41 | 78.76 | 78.86 | 78.82 | 78.18 | 78.54 | 78.72 | 78.76 | 78.78 |
| | Edge Set | 72.17 | 71.64 | 72.06 | 72.20 | 71.93 | 72.11 | 72.01 | 72.27 | 72.01 |

Table 15: Shortest path MSE on all connected graphs with 8 nodes. We highlight the best performance per training task in columns.

| | | Original GraphToken Encoder Training Task: | | | | | | | | |
|---|---|---|---|---|---|---|---|---|---|---|
| | **Method** | **Node count** | **Edge count** | **Cycle check** | **Triangle counting** | **Node degree** | **Connected nodes** | **Reachability** | **Edge existence** | **Shortest path** |
| Non-linear | GCN | 2.27 | 2.24 | 2.31 | 6.07 | 6.07 | 2.06 | 6.07 | 11.09 | 3.75 |
| | GIN | 2.57 | 2.77 | 2.83 | 2.93 | 2.52 | 2.54 | 4.84 | 3.09 | 3.61 |
| | MPNN | **0.29** | **0.29** | **0.76** | **0.31** | **0.71** | **0.49** | **0.75** | 1.58 | **0.51** |
| | HGT | 3.03 | 2.64 | 2.27 | 1.60 | 2.60 | 2.14 | 1.80 | 1.95 | 2.81 |
| | MHA | 3.04 | 0.71 | 0.95 | 0.78 | 1.01 | 0.74 | 1.74 | **1.55** | 1.05 |
| Linear | Node Set | 2.35 | 2.35 | 2.35 | 2.36 | 2.36 | 2.35 | 2.34 | 2.36 | 2.34 |
| | Edge Set | 2.99 | 2.99 | 2.99 | 2.99 | 2.97 | 2.97 | 2.99 | 2.99 | 2.99 |

Table 16: # of triangles MSE on all connected graphs with 8 nodes. We highlight the best performance per training task in columns.

| | Method | Node count | Edge count | Cycle check | Triangle counting | Node degree | Connected nodes | Reachability | Edge existence | Shortest path |
|---|---|---|---|---|---|---|---|---|---|---|
| Non-linear | GCN | 132.94 | 129.03 | 164.53 | 316.07 | 316.07 | 127.17 | 316.07 | 690.03 | 293.53 |
| | GIN | 152.13 | 168.35 | 182.95 | 201.64 | 169.71 | 156.16 | 251.23 | 200.45 | 251.65 |
| | MPNN | **8.33** | **7.51** | **32.08** | **4.56** | **51.90** | **27.18** | **51.04** | 124.89 | **41.73** |
| | HGT | 191.14 | 170.71 | 165.88 | 126.92 | 172.84 | 160.29 | 156.10 | 136.22 | 175.45 |
| | MHA | 197.36 | 30.27 | 96.56 | 27.10 | 59.58 | 52.42 | 138.48 | **80.22** | 60.72 |
| Linear | Node Set | 167.81 | 168.72 | 167.33 | 167.40 | 167.90 | 167.96 | 168.57 | 169.38 | 166.13 |
| | Edge Set | 181.44 | 181.21 | 181.18 | 181.32 | 180.86 | 179.44 | 181.08 | 181.68 | 181.40 |

Table 17: Degree Accuracy on all trees with 15 nodes. We highlight the best performance per training task in columns.

| | Method | Node count | Edge count | Cycle check | Triangle counting | Node degree | Connected nodes | Reachability | Edge existence | Shortest path |
|---|---|---|---|---|---|---|---|---|---|---|
| Non-linear | GCN | 53.57 | 55.15 | 55.24 | 25.91 | 25.91 | 54.86 | 25.91 | 11.08 | 36.51 |
| | GIN | 60.35 | 58.79 | 56.36 | 55.11 | 59.88 | 68.04 | 42.01 | 66.72 | 55.25 |
| | MPNN | **79.37** | **78.36** | 59.18 | **72.35** | 62.38 | 65.90 | 57.37 | 57.33 | 58.45 |
| | HGT | 54.88 | 55.33 | 55.34 | 58.65 | 54.33 | 58.84 | 57.27 | 57.43 | 55.34 |
| | MHA | 59.17 | 61.61 | 60.38 | 57.18 | 54.99 | 61.00 | 52.29 | 58.56 | 53.95 |
| Linear | Node Set | 65.64 | 66.32 | 65.93 | 66.10 | 66.13 | 65.95 | 66.28 | 66.22 | 65.82 |
| | Edge Set | 69.59 | 69.87 | **69.44** | 69.40 | **69.86** | **69.56** | **69.32** | **69.55** | **69.66** |

Table 18: Diameter Accuracy on all trees with 15 nodes. We highlight the best performance per training task in columns.

| | Method | Node count | Edge count | Cycle check | Triangle counting | Node degree | Connected nodes | Reachability | Edge existence | Shortest path |
|---|---|---|---|---|---|---|---|---|---|---|
| Non-linear | GCN | 50.77 | 50.36 | 49.54 | 25.97 | 25.97 | 50.01 | 25.97 | 6.77 | 26.64 |
| | GIN | 58.29 | 54.44 | 52.24 | 49.41 | 51.47 | 59.62 | 24.11 | 58.77 | 46.27 |
| | MPNN | 54.24 | 54.68 | 54.97 | 59.29 | **67.65** | **63.80** | 54.13 | 52.05 | 59.48 |
| | HGT | 57.15 | 54.88 | 54.90 | 57.58 | 57.05 | 65.22 | 54.51 | 58.70 | 53.07 |
| | MHA | 53.95 | 56.63 | 60.41 | 54.62 | 53.39 | 56.07 | 52.85 | 55.17 | 51.70 |
| Linear | Node Set | **61.89** | **62.68** | **62.74** | **62.36** | 61.99 | 61.93 | **62.34** | **62.49** | **62.40** |
| | Edge Set | 56.57 | 56.19 | 56.27 | 56.83 | 56.25 | 56.53 | 56.31 | 56.72 | 56.84 |

Table 19: Shortest path MSE on all trees with 15 nodes. We highlight the best performance per training task in columns.

| | Method | Node count | Edge count | Cycle check | Triangle counting | Node degree | Connected nodes | Reachability | Edge existence | Shortest path |
|---|---|---|---|---|---|---|---|---|---|---|
| Non-linear | GCN | 12.95 | 12.31 | 12.62 | 26.17 | 26.17 | 12.22 | 26.17 | 49.78 | 21.71 |
| | GIN | 9.57 | 10.69 | 11.32 | 11.88 | 11.03 | 8.37 | 19.35 | 9.76 | 14.39 |
| | MPNN | **4.19** | **4.54** | 9.82 | **4.92** | **6.87** | **6.10** | 11.06 | 12.10 | 11.01 |
| | HGT | 10.57 | 10.96 | 11.65 | 9.09 | 12.56 | 8.17 | 10.76 | **9.26** | 10.98 |
| | MHA | 10.49 | 9.88 | **9.51** | 11.22 | 12.75 | 10.52 | 13.31 | 10.09 | 12.78 |
| Linear | Node Set | 10.20 | 10.05 | 10.13 | 10.11 | 10.17 | 10.21 | 10.07 | 10.18 | 10.03 |
| | Edge Set | 9.92 | 9.87 | 9.92 | 9.93 | 9.88 | 9.88 | **10.01** | 9.91 | **9.87** |

