# OpenReview forum: "Parameter Efficient Graph Encoding for Large Language Models"
_ICLR.cc/2025/Conference — Submitted to ICLR 2025_

### Official Review · Reviewer_Zq2s · 2024-10-24

**Soundness:** 2
**Presentation:** 3
**Contribution:** 2
**Rating:** 6
**Confidence:** 3

**Summary:**

The paper introduces GraphToken, a novel parameter-efficient approach for encoding graph data and solving graph reasoning tasks with LLMs. GraphToken leverages GNNs to encode graph information into continuous encodings, which contain explicit structural information. Such encodings are passed to LLMs to handle graph reasoning tasks for node, edge, and graph levels. The GraphToken method retains frozen LLM parameters, which reduces computational requirements compared to other Graph + LLM work. The experimental results on GraphQA and molecular property prediction from ChemLLMBench indicate that smaller LLMs equipped with GraphToken can outperform much larger models.

**Strengths:**

- The use of GNNs to encode graph structures into LLM-compatible tokens is a good contribution, as it addresses a gap in the current approaches of LLMs for graph data.

- The experiments cover a wide range of models and tasks, with different LLMs and GNN architectures, also evaluated on node, edge, and graph-level reasoning tasks, and chemical property prediction benchmarks.

- The ability to boost smaller LLMs to surpass larger ones has important implications for resource-efficient ML and real-world deployment, which may also contribute to LLM reasoning study.

- The paper is clearly written and easy to follow.

**Weaknesses:**

- While the paper shows strong results, a more detailed analysis of where GraphToken underperforms would strengthen the contribution. For example, it would be helpful to know if there are specific graph types or structures where the method struggles. Maybe provide a breakdown of performance across different graph sizes or structures, or to include error analysis on specific examples where GraphToken struggles.

- One limitation of the method is that it requires access to an open-source LLM to compute gradients. This could be a challenge for 1) the stronger proprietary LLMs have no such access, limiting the method’s applicability in certain contexts. 2) computing gradients for larger models can be expensive, even though the LLM parameters are not tuned.

- Adding a more explicit discussion of limitations can strengthen the paper. For example, generalization to larger or more complex graphs.

**Questions:**

- The results show that larger models have an advantage over the proposed GraphToken + small model for the connected nodes task. Are there any insights or analysis about why this is the case? Why do larger models perform particularly well for this task? Can you provide a more detailed analysis of the connected nodes task, perhaps including visualizations or examples that illustrate why this task might be particularly challenging for GraphToken.

- The method is claimed to be parameter efficient. Any runtime analysis or comparison to the baselines?

---

> ### Author Response · Authors · 2024-11-18
> **Rebuttal**
>
> Thank you for your thoughtful review!  We briefly address some of the questions and comments you raised:
>
> ### **Why can a larger model do better on the connected nodes task?**
>
> Great question!!  Actually, very recent theoretical results from [1] provide insight to this behavior, providing bounds for the kinds of tasks where Transformers can outperform (non-attention) GNNs (and vice-versa).  Their results show that Transformers have an advantage on tasks which require access to the full graph in order to answer them (like connected nodes, and shortest path).  It's an open research question on whether a new kind of GNN encoder can do better than Transformer-based ones in this area.
>
> We will happily add a discussion on the connected nodes task, this related theory, and the implications for encoder design to this work!
>
> ###   **Any runtime analysis or comparison to the baselines for Parameter Efficiency?**
>
> Another great question -- here is a runtime analysis for the PeFT experiments in Table 3!!
>
> |  |  | Trainable parameters | BACE | BBBP | ClinTox\_2 |
> | :---- | :---- | ----- | ----- | ----- | ----- |
> | Model | Method |  | Training Hours | Training Hours | Training Hours |
> | Gemma-2B | Prompt tuning | 40960 | 1.55 | 2.14 | 1.49 |
> | Phi2-2.7B | Prompt tuning | 51200 | 1.63 | 2.27 | 1.71 |
> | Mistral-7B v0.3 | Prompt tuning | 81920 | 3.98 | 3.97 | 3.99 |
> | Gemma-2 2B | LoRA finetuning (input embed tuning) | 516096 | 0.30 | 0.47 | 0.28 |
> | Phi3.5 3.8B | LoRA finetuning (input embed tuning) | 70272 | 0.37 | 0.98 | 0.57 |
> | Mistral-7B v0.3 | LoRA finetuning (input embed tuning) | 73728 | 0.77 | 1.07 | 0.76 |
> | Gemma-2 2B | P Tuning (1 token, 100k params) | 97208 | 13.92 | 13.90 | 13.92 |
> | Phi3.5 3.8B | P Tuning (1 token, 100k params) | 129464 | 22.37 | 18.03 | 15.42 |
> | Mistral-7B v0.3 | P Tuning (1 token, 100k params) | 172472 | 23.95 | 23.89 | 23.87 |
> | **Gemma-2 2B**| **ChemGraphToken (MPNN)** | 299520 | **0.13**| **0.27** | **0.19** |
>
> This Table shows that GraphToken is not only the most performant model for the task, but also the fastest!  (This is by design -- GNN parameters are ‘computationally cheaper’ than Transformer parameters).  Please see our discussion with who had some similar questions about efficiency.
>
> We have a longer discussion on the efficiency of the method in a dedicated comment to Reviewer #AQrB, that highlights.
> 1. **Runtime efficiency** (shown above)
> 2. **Asymptotic complexity** benefits (no big surprises, but a MPNN is more efficient than a Transformer)
> 3. More about the *parameter usage benefits* (where we see adding $O(100k)$ parameters allows a small model (PaLM-2-S) to beat a substantially larger model (PaLM-2-L).
>
> ### **GraphToken Losses and Limitations**
>
> Great idea!  We’re working on a more detailed analysis of where GraphToken under performs, but the analysis is taking time.  We hope to complete it this week!
>
> ### **Requires access to an open-source LLM to compute gradients**
>
> We completely agree that the current inability to compute gradients for proprietary LLMs may prevent the application of GraphToken to closed-source models.  However we would like to emphasize a few points:
>
> 1. GraphToken makes small models ‘punch above their weight’.  This can really enable new capabilities for smaller open source models.  For example, we have a nice experiment comparing GraphToken to Tx-LLM, a specialized text model for therapeutics for Reviewer #FJUo.  We see that a smaller (and more general model) can outperform a more specialized model, even with fine tuning on the same task!
> We believe this to be a significant contribution.
>
> 1. As strong open source LLMs continue to be released, GraphToken can be applied with open models which have 100 billion+ parameters.
>
> 1. Finally, we believe that the strong performance on algorithm reasoning tasks will motivate developers of proprietary LLMs to provide support for graphs and other modalities.
>
> ### **References**
>
> [1] “Understanding Transformer Reasoning Capabilities via Graph Algorithms” - https://arxiv.org/pdf/2405.18512

---

> > ### Author Response · Authors · 2024-11-22
> > **Limitations discussion forthcoming**
> >
> > > GraphToken Losses and Limitations
> >
> > We just wanted to send a brief reply that we have finished extensive experimentation (including multi-task generalization) requested by other reviewers and will be turning our attention to the limitations discussion you requested.
> >
> > We apologize for the delay and thank you for you patience!

---

> ### Author Response · Authors · 2024-11-25
> **Limitations Discussion and Graph Structure Analysis**
>
> Thanks for your question on the limitations of GraphToken -- we have analyzed its performance across graphs structures and will add the following discussion on it to the manuscript!
>
> ### **Limitations of GraphToken**
>
> Our results have shown that GraphToken is both a flexible and generalizable encoder of graph structured data for LLMs. Here we discuss some limitations of the method as inspiration for future work.
>
> ### Encoder Generalizability
>
> The main limitation of GraphToken is that its encoder might learn spurious correlations due to idiosyncrasies in the distribution of input graphs it was trained on.  As such, it's important that the encoder works robustly regardless if its evaluated on a different distribution of input graphs (w.r.t. their density, number of nodes/edges, etc).  We note that this is a general weakness of GNNs and not specific to GraphToken itself.  As such, there is a rich literature on creating robust GNNs [1,2] that has made significant progress in creating more generalizable GNN architectures.  We expect that these results will be directly able to be “plugged in” to GraphToken encoders and will greatly aid in their generalization.
>
> ### **Does Graph Structure Matter?**
>
> It is interesting to study how graph structure might affect a GraphToken encoder’s performance on downstream tasks.  Are more complicated graphs harder?  Do other structural patterns influence its results?
>
> #### Experiment design:
>
> We use the Gemma2-2B multi-task encoder we trained for **reviewer #FJUo** which is trained on 9 out of 10 GraphQA tasks.
> Then we calculated a number of graph properties that have been shown useful for analyzing GNN performance [1], and examined the correlation between these graph characteristics and whether the model was able to correctly answer its task (out of 10 GraphQA tasks) on unseen graphs.  The results are as follows:
>
> | Graph Property | Pearson correlation |
> | :---- | :----: |
> | num\_nodes | \-0.146 |
> | num\_edges | \-0.0581 |
> | edge\_density | 0.0732 |
> | degree\_gini | \-0.104 |
> | avg\_degree | \-0.040 |
> | avg\_cc | 0.0111 |
> | transitivity | 0.0103 |
> | num\_triangles | 0.000120 |
> | cc\_size | \-0.0344 |
>
> Interestingly, we find that most graph properties are uncorrelated with the downstream task’s performance.  We do see a weak negative correlation between the number of nodes in the graph and correctly answering tasks.  However this is expected -- as the number of nodes grows, the graph has the potential for more complexity.
>
> We believe that **this result supports the strong generalization capabilities of GraphToken**!
>
>
> #### **References**
>
> [1]  GraphWorld: Fake Graphs Bring Real Insights for GNNs -  https://arxiv.org/abs/2203.00112
>
> [2]  A Survey of Deep Graph Learning under Distribution Shifts: from Graph Out-of-Distribution Generalization to Adaptation - https://arxiv.org/abs/2410.19265v1
>
> [3] Robustness of Deep Learning Models on Graphs: A Survey - https://galina0217.github.io/works/robust_survey.pdf

---

> > ### Comment · Reviewer_Zq2s · 2024-11-26
> >
> > Thank the authors for their response. I have no more questions, and I will keep my score.

---

### Official Review · Reviewer_wXu1 · 2024-10-29

**Soundness:** 3
**Presentation:** 4
**Contribution:** 3
**Rating:** 6
**Confidence:** 4

**Summary:**

This paper presents a novel approach to integrating graph information into frozen large language models (LLMs). Given recent findings that graph serialization may be inefficient, the authors propose encoding the graph through a specialized neural network, whose output is then fed into the LLM via soft prompting/prefix tuning.

**Strengths:**

The manuscript is well-organized and clearly written, making it easy to follow. The experimental section is robust, featuring evaluations on two diverse dataset collections—GraphQA and ChemLLMBench—as well as a detailed ablation study exploring the impact of different graph encoders.

The proposed model shows significant performance gains over traditional graph serialization techniques, utilizing a relatively compact graph encoder with around 1-2 million parameters.

Overall, the contributions in this paper are significant as it is one of the first (if not the first) works that use a graph encoder in adjacency with a frozen LLM, and addressing the concerns could further solidify its impact.

**Weaknesses:**

1. Comparisons with GraphLLM: In line 156, the authors reference another graph-encoder-based approach (GraphLLM) suggesting it is limited to specific graph types. However, GraphLLM claims compatibility with simple graphs, such as those in GraphQA, and can also handle graphs containing additional textual information. In contrast, GraphToken may not accommodate such graph structures. This distinction should either be clarified in the text or, if I have misunderstood, an experimental comparison with GraphLLM could help elucidate the model’s advantages more directly.

2. Task-Dependent Encoder Output: Line 202 notes that the graph encoder’s output is tailored to the specific downstream task and there is no universal approach.
- First, this could be considered a form of data leakage. To address this, the experimental section should at least include a performance evaluation for tasks such as “node degree” utilizing a single token (i.e., the same graph encoder for all tasks).
- Additionally, the fact that, in some cases, a single token per node/edge is utilized raises questions about the scalability to larger graphs (graphs with significantly more nodes than ~12 which is the average size in  GraphQA). Has the model been evaluated with larger graphs? Insights into its performance under these conditions would be valuable.
- Lastly, such as reliance on the downstream task raises some concerns around how the model could be actually deployed as a general-purpose Large Graph+Language Model. Further clarification on this point would strengthen the paper’s contributions.

**Questions:**

In Section 7, it is noted that experiments were conducted on TPUs; however, details on the model’s memory requirements are not provided.

---

> ### Author Response · Authors · 2024-11-18
>
> Thank you for your time and careful consideration of our work!  We want to address your concerns raised in the Weaknesses section:
>
> ### **GraphToken Features**
>
> As GraphToken uses a general GNN encoder, it accommodates any featurization of the nodes, edges, or graph including textual information. An example of this capability is demonstrated in section 4.2 where we consume molecular data which includes very rich information -- including node, edge, and graph level features. We do not compare to GraphLLM because, to quote the GraphLLM paper “In this paper, we operate under the assumption that nodes (or entities) can be characterized by textual features”.  (We have actually tried to compare to GraphLLM and some similar works regardless, but have not succeeded in reproducing them -- in our experience so far, papers with significant architecture modifications seem more brittle.)
>
>
> ### **Task-Dependent Encoder Output**
> 1. We do not consider this data leakage. Instead we consider this part of the prompt.  In the parameter efficient fine-tuning setting, task performance is usually related to fine-tuning a model for a specific task.  (Practically speaking, a retrieval system designed to use this graph information could easily select a graph-encoder which would be specifically trained for its task.)
>
> 1. While we do indeed experiment on smaller graphs, this is primarily just to benchmark against text encoding in LLMs which have very small context windows.  One benefit of this approach is that it is significantly more scalable than alternative text encoding formulations which may require hundreds of tokens to encode the information about these graphs that we are compressing into a single token.
>
> 1. While we consider the parameter-efficient fine-tuning setting (improving a model on one graph task), we do agree with you that this is an interesting direction.  One benefit of our design is that we can easily accommodate advances in graph encoders (either specific ones like better molecular encoders, or general graph reasoning architectures).  The encoders can also be pretrained on their tasks (independent of the LLM).  We believe our results show the promise of this design (utilizing graph soft tokens) and that this path will allow us to add fundamental new capabilities to large language models.
>
> ### **Model memory requirements**
>
> Great question!  The biggest requirement of our method is computing gradients from a LLM.  While this does not necessarily require loading the entire model into memory, doing so definitely decreases the training latency.
>
> - Our experiments with GraphToken + Gemma 2B were run on a single machine with one TPU v5e (16GB VRAM).
>
> - The experiments with GraphToken + PaLM 2-S were run using TPU v3 accelerators. Unfortunately, we are not able to provide more detailed accelerator information for PaLM 2-S (as that is strongly related to model size).
>
> - The GNNs (with only 100k parameters) don’t require much additional VRAM headroom over the LLM requirements.
>
> Finally, we note that the memory requirements can be pushed substantially lower using standard techniques for minimizing LLM memory use such as quantization, or by loading only part of the model into memory at a time.

---

> ### Author Response · Authors · 2024-11-27
> **Effect of Graph Size**
>
> We apologize for the late reply about this concern!
>
> > Additionally, the fact that, in some cases, a single token per node/edge is utilized raises questions about the scalability to larger graphs (graphs with significantly more nodes than ~12 which is the average size in GraphQA). Has the model been evaluated with larger graphs? Insights into its performance under these conditions would be valuable.
>
> This is a great question.
> - In the soft prompt tuning literature, the number of tokens to generate is a typically a hyper-parameter which can be varied to increase model capacity.
> - We **do not** advise using just one GraphToken for larger and more complex graphs!
> - However, to be frank, we have been impressed with how well just one token can do -- the multitask experiment for Reviewer FJUo uses one GraphToken and can perform the 9 tasks it was trained on, in addition to an unseen 10th task!
>
> To answer the related question of *'How much work could one GraphToken do?'* -- we can produce a larger version of GraphQA (same tasks, but with larger generated graphs) and perform some analysis on how answer quality degrades as graph size grows.
>
> Finally, we call your attention to the correlation experiment for Reviewer Zq2s, where we see that larger graph size (#nodes) is only weakly correlated with incorrect answers.  However, we agree with your intuition that one should only be able to fit 'so much' graph inside a single token.
>
> Thanks and please let us know if you have additional concerns or would like this additional experimentation.  (it would not be too hard to do)

---

### Official Review · Reviewer_FJUo · 2024-10-29

**Soundness:** 2
**Presentation:** 3
**Contribution:** 2
**Rating:** 6
**Confidence:** 4

**Summary:**

This paper proposes a method to combine graph models and LLMs to query graph structural properties better. It uses a graph encoder, such as a GNN or a graph transformer, to embed a graph into graph tokens and combine the graph tokens with text/question tokens as the input to a downstream LLM, for generation. Various experiments are conducted, and the method shows strong improvement over existing baselines on graph and molecule properties prediction tasks.

**Strengths:**

- The idea of the paper is clearly conveyed, and the writing is easy to follow.

- While most existing GNN+LLM work focuses on text-attributed graph learning ability, this paper addresses an important issue on querying graph structures.

**Weaknesses:**

- While I agree graph property querying is crucial to align graph information with LLM, the presented work is only weakly connected to LLMs. In this work, the LLM merely serves as a decoder that answers "yes" or "no" based on the embeddings generated from a graph encoder, and it is not much different from an MLP classifier. The goal of incorporating an LLM is to use its generalizability and human-interpretability. Take your generalization study as an example, a meaningful integration with LLM is when you train the model with triangles and cycle check tasks, you can directly use the model to tackle bipartite check tasks by only changing the question to "Is the graph bipartite?" Can you justify why the integration of an LLM is necessary when your model seems to be a supervised-task-only model?

- While the paper is from a graph structure querying objective, similar architectures are proposed by many existing (though possibly contemporary works). Just to name a few approaches that encode graphs with a graph encoder and align that with an LLM [1,2,3,4]. The discussion of this line of work is missing from the paper, and I think the connection of the proposed work to PEFT is relatively weak.

- The learning task design is over-simplified and can be easily captured by a graph model or position encodings. It is like including the answer in the question, but only in an implicit way. If I understand correctly, all tasks are trained separately, and the model cannot learn from similar tasks (cycle check and triangle count can help each other). I suggest a discussion on why these tasks are helpful and meaningful and why not incorporate more tasks and train them together to build a large graph property prediction model.

- While the proposed approach outperforms LLM baselines on the molecule property prediction tasks, it actually underperforms graph models (the authors should consider adding graph models into baseline comparison). This relates back to my first weakness, that is, the performance boost seems to be solely attributed to the use of a stronger graph model and has very little to do with an LLM. To show that LLM is necessary, you should consider providing results that show that adding LLM improves the performance over the "graph-encoder-only" variant, or, the model directly generalizes to unseen molecule targets.

[1] Tang, Jiabin, et al. "Graphgpt: Graph instruction tuning for large language models." Proceedings of the 47th International ACM SIGIR Conference on Research and Development in Information Retrieval. 2024.

[2] Kong, Lecheng, et al. "Gofa: A generative one-for-all model for joint graph language modeling." arXiv preprint arXiv:2407.09709 (2024).

[3] Chen, Runjin, et al. "Llaga: Large language and graph assistant." arXiv preprint arXiv:2402.08170 (2024).

[4] He, Yufei, and Bryan Hooi. "UniGraph: Learning a Cross-Domain Graph Foundation Model From Natural Language." arXiv preprint arXiv:2402.13630 (2024).

**Questions:**

Please see weaknesses

---

> ### Author Response · Authors · 2024-11-18
>
> We thank the reviewer for a very insightful response!
>
> ### **Scope of Work**
> We believe that while the overall direction the reviewer highlighted is in line with our work, the scope of our work is in parameter-efficient encoding. There is more that LLMs bring besides generalizability and interpretability – it’s their in-context learning capabilities [1]. When considering extremely significant improvements of GraphToken over LLM-only baselines (cf. **Palm-2 L** results in Table 1), it is clear that the GNN encoder is a better overall design of the system.  In fact, we would say the core argument of our paper is that “GNNs are necessary for LLM advancement".
>
> We believe the design ideas sketched in this review are more general, but our work solves a particular encoding problem for graphs that has not been properly addressed before.
>
> ### **Unified Model**
> To the point of training a unified model, it is certainly a useful experiment; however, in the parameter-efficient fine-tuning literature it is common to fine-tune LLMs to perform a single task that is most useful to the concrete goal the user has. We take our graph task definitions from Fatemi et al. (2023) [2], which is a standard for this fast-evolving field. While simple, they span different levels of complexity for both transformers and GNNs – for instance, GNNs alone can not solve the connectivity task without resorting to global positional encoding.
>
> ### **Comparison with Specialized LLMs**
> To put our results on property prediction in context, we compare with Tx-LLM [3], a special purpose LLM designed for medical applications (with a special emphasis on biochemical interactions).   Both models are fine tuned for the task.  Please note that Tx-LLM is “text-only” but has significant pre and post training for the domain.  GraphToken received no additional task-specific pretraining or fine tuning.
>
> | model | ClinTox Accuracy |
> | ---- | ---- |
> |GraphToken Gemma-2B (MPNN) | **0.879** |
> |Tx-LLM Gemma-7B | 0.851 |
>
> We see that GraphToken is again able to outperform a larger model (even though it's significantly more specialized).
>
>
> ### **Better GNN models available?**
> We completely agree with you that better GNNs for molecular representation should improve these results further -- on both specialized tasks (and grounding/generation for explainability).   We view this as a positive statement about our architecture’s benefits -- it allows LLMs to leverage the strengths of domain-specific models.
>
>
> ### **Related Work**
> Thank you for the references, we are happy to extend the related work with a discussion of these papers.  We believe that the rapid growth of contemporary work in the area justifies the timeliness and relevance of our work!
>
> ### **Why use a GNN to create LLM input?**
>
> At the core, converting from an unordered object (a graph) into a sequence introduces a bias that is hard to develop heuristics for [4].  The primary benefit from our proposed architecture is that the graph encoder can preserve properties (like equivariance, etc) for as long as they are useful, and then deliver an optimized representation for the LLM.  This allows graph data to be used just like “another modality” (vision, audio, etc).
>
> ### **References**
>
> [1] “Language Models are Few-Shot Learners” - https://arxiv.org/pdf/2005.14165
>
> [2] "Talk like a graph: Encoding graphs for large language models" - https://arxiv.org/pdf/2310.04560
>
> [3] “Tx-LLM: A Large Language Model for Therapeutics” - https://arxiv.org/pdf/2406.06316
>
> [4] "Towards an Understanding of Graph Sequence Models" - https://openreview.net/pdf?id=iaHghgG8NR

---

> > ### Comment · Reviewer_FJUo · 2024-11-21
> >
> > Thank you for your reply. While the authors address some of the concerns, my main concerns about the paper remain.
> >
> > # Scope of work
> >
> > The mentioned in-context ability is, in fact, my concern. As I suggested in the original review,
> >
> > > "a meaningful integration with LLM is when you train the model with triangles and cycle check tasks, you can directly use the model to tackle bipartite check tasks by only changing the question to 'Is the graph bipartite?'"
> >
> > This is exactly the in-context learning setting. Then my question is, as you integrate LLM into the framework, does your model have in-context learning ability? And, do you have an experiment to show such in-context learning ability? This is my main concern, and if the answer to both questions is yes, the contribution of the paper will be greatly improved and I am prone to acceptance. The comparison to Palm-2 L does not really say anything, Palm-2 L is in-context, while PaLM-2-S with GraphToken is not.
> >
> > # Better GNN
> >
> > I apologize if my original review is not clear. My concern here is that all experiments are purely supervised, so you should at least compare GraphToken to traditional GNN methods, there are many existing baselines.
> >
> > Lastly, I totally agree that a graph encoder is necessary for learning (potentially with LLM) when dealing with graph data, but this paper is not the first one proposing this argument. More importantly, the integration at the current stage completely sacrifices the in-context learning, generalizability, and zero-shot learning ability of LLM, and does not really advance LLM.

---

> ### Author Response · Authors · 2024-11-22
> **Response**
>
> Thanks for the reply!  In the meantime, we have been **very actively** working to answer the questions you raised in the original review!
>
> ## **MultiTask: Yes!**
>
> To illustrate that the same GraphToken can work for multiple tasks, we have investigated multi-task models at your request.  We trained a three task model (edge existence, cycle check, reachability) with Gemma 2B which yielded the following results:
>
> | Task | MultiTask Model Accuracy | Single Task Accuracy |
> | :---- | :---- | :---- |
> | Cycle Check | 0.986 | 0.988 |
> | Edge Existence | 0.652 | 0.690 |
> | Reachability | 0.932 | 0.932 |
>
> We see a small drop in performance from using a single encoder, but are quite pleased with the results.  (The multi-task model has not had its hyper-parameters optimized, so there is likely more performance available).
>
> #### **Generality**
> We emphasize that this model didn’t have access to the task description (exactly the same GraphToken representation was being provided for each of the three tasks).  We believe that this shows the generality of the approach.
>
> #### **More tasks?**
> We have begun training even larger multitask models (i.e. with more tasks), but hit some infrastructure issues along the way and the experiments are not done yet.  We may have additional results to report prior to the rebuttal prior ending.
>
> While these initial results seem quite promising, we suspect that training a true “graph foundation model” is a research work unto itself.  For example, perhaps some tasks can not be easily combined.  We note that multi-task performance on current GFM models seems to be lacking [1].
>
>
> ## **GNN-only Results**
>
> We agree that these are valuable both to illustrate what headroom there might be for the combined GNN + LLM architecture, and further the field.
>
>
> ### **GraphQA GNN-Only Results**
>
> We have run the following experiments which we will add to the paper:
>
> For each task, we ran the following additional baselines:
> - a “GNN-only” architecture using the GraphToken encoder
> - Gemini 1.5 Pro (a recent capable “LLM-only” model)
>
> | Task | GNN-Only | LLM-Only |
> | :---- | :---: | :---: |
> | Reachability (connectivity) | 93.6 | **95.0** |
> | Connected Nodes | **96.8** | 69.0 |
> | Shortest path | **70.2** | 47.0 |
> | Edge existence | 72.2 | **93.2** |
> | Node count | **100** | 99.4 |
> | Edge count | **87.4** | 43.6 |
> | Cycle check | **98.8** | 95.4 |
> | Triangle counting | **39.4** | 18.2 |
> | Node degree | **99.4** | 60.6 |
>
> Our intent is to use these experimental results to add a discussion of the tradeoffs between GNN, GNN + LLM, and LLM architectures to the paper, informed by recent theoretical results in the area.
>
>
> ### **GNN-Only Molecular Property Baselines**
>
> We also investigated running GNN-only molecule encoders, but have so far found these encoders difficult to adapt to our programming environment.  We hope that the GraphQA results illustrate the strength of GNNs on the task to your satisfaction.
> If not, we are happy to report other’s numerical results on the molecular property prediction datasets if you wish.
>
> ### **Minor points**
>
> Finally we wanted to respond to two minor points that we believe came up in your review.
>
> #### **GNN + LLM benefit**
>
> We are actually aware of some follow up work (which cites us) that demonstrates a unique benefit for our combined GNN + LLM architecture in a few-shot learning setting.  In this setting, the GNN+LLM is better able to adapt to new unseen tasks.
>
> #### **Novelty**
>
> Similarly, we are fairly certain that we were very early (perhaps the first) advocates of this architecture.  However we acknowledge that we are not the first to get published on it.
>
> *Unfortunately we can not provide this citation (or evidence of our novelty) for reasons of anonymity.*   We will leave a note for the AC about this, but suspect there is no good solution.
>
> ### **Thanks!**
>
> Thanks for your time spent in the review process and engaging with the authors!  Please let us know if there are remaining points we have failed to address.
>
>
> ### **References**
>
> [1] Text-space Graph Foundation Models: Comprehensive Benchmarks and New Insights - https://arxiv.org/pdf/2406.10727

---

> ### Author Response · Authors · 2024-11-22
> **MultiTask Generalization: ✅**
>
> ### **MultiTask Generalization: Yes!**
>
> In order to illustrate MultiTask generalization, we have performed the following experiment at your request:
>
> -  We have trained a MultiTask GraphToken model on 9 of the 10 GraphQA tasks.  The LLM backbone is Gemma2-2B (to minimize training time).
> -  As before the same encoder is used for all tasks, and the encoder has no knowledge about the task while encoding.
> -  We evaluate this model on a *withheld* task -- cycle check.  It has never seen cycle check before this eval.
>
> ### Results:
>
> For context, we provide the single task performance ("GraphToken Single Task") on Gemma2-2B as well as its text-only performance.
>
> |  | MultiTask GraphToken (Generalization) | GraphToken Single Task |  Gemma2-2B (text only)
> | :---- | :----: | :----: |  :----: |
> | Cycle Check (Accuracy) | 88.4 | 98.8 | 60.0 |
>
> We see that while there is a drop on performance (as might be expected) compared to optimizing a task directly:
> 1.  **GraphToken can generalize to a unseen task**
> 2.  **The generalization outperforms large models (e.g. PaLM-2-L) on the task.**
>      1.  PaLM-2-L's highest scoring result was 83.3.
>      2.  The generalization is much better than using a text representation of the graph with the backbone LLM.
>
> We thank you for the suggestion and will be including this experiment in an updated version of the paper!
>
> We believe that this is a strong rebuttal to the concerns from your original review.  Please let us know if you have any additional questions.

---

> > ### Comment · Reviewer_FJUo · 2024-11-26
> >
> > Thank you again for the response. I think the updated results are very interesting and indeed resolve my concern. I am leaning towards acceptance, and have updated my score. My last question is that the leave-out dataset (cycle check) seems arbitrary, have you tried all leave-out settings (leave shortest-path out, etc)? If other settings do not work, it does not change the quality of the paper, and the current cycle check results are significant enough, I just think it seems like a natural and easy extension, but a strong justification. Please don't take this as a request for more experiments, I am already impressed by your diligence, and you can work on this in future revisions.

---

> ### Author Response · Authors · 2024-11-26
>
> >  the leave-out dataset (cycle check) seems arbitrary, have you tried all leave-out settings (leave shortest-path out, etc)?
>
> Great question - this was the first task train/test split we tried actually.  We choose cycle check as a test because it seemed to perform well in the three task experiment above.
>
> We'd like to extend the analysis further for the camera ready, and we agree that a full leave-one-out cross-validation would be very interesting.  Unfortunately we ran into some resource issues that prohibited us from pursuing this question more this week.
>
> Thanks for the suggestion and engaging in the review process!

---

### Official Review · Reviewer_AQrB · 2024-11-03

**Soundness:** 2
**Presentation:** 2
**Contribution:** 2
**Rating:** 5
**Confidence:** 3

**Summary:**

This submission addresses the interesting problem of using large language models (LLMs) for graph reasoning tasks. To transform structured graph data into a sequential form suitable for LLMs, the authors propose a parameter-efficient method called GraphToken. This approach leverages graph neural networks to learn graph embeddings, which are then concatenated with text embeddings to form the inputs for LLMs. However, the paper could benefit from a clearer clarification of the motivation and technical contributions, as well as the inclusion of additional baselines and experimental results.

**Strengths:**

S1. This submission presents an interesting solution to integrate the graph reasoning task with LLMs.

S2. The presentation is clear and easy to follow.

S3. The empirical results demonstrated by authors significantly outperform baselines methods.

**Weaknesses:**

**Weakness**

W1. The motivation is not well-explained.

W2. The technical contributions are insufficiently detailed.

W3. Important literature on graph foundation models is missing.

W4. There is a lack of efficiency comparisons.


**Concerns**

C1. In the motivation, the paper mainly criticizes prior graph foundation models that use text descriptions as input for graphs, leading to the claim: “This highlights the need to explore better and more expressive ways of representing structured data to an LLM.” However, the authors overlook the fact that previous graph foundation models were built to integrate seamlessly with any open-source or closed-source LLM, aiming to externally drive LLMs to perform graph reasoning tasks. In other words, this should highlight the need to explore better and more expressive ways of representing structured data to an LLM **in a convenient and flexible way**. The graph representation and embedding fusion method proposed in this paper may not achieve this goal.

C2. From a technical perspective, this submission employs a classical mechanism of embedding fusion using a graph encoder and a text encoder. The authors need to clarify what the technical contributions of this mechanism are.

C2-1. To present the technical contributions, the authors could elaborate on the core issue: “There is often no clear choice in what order to sequentially write the graph data.” The authors are expected to explain what research challenges this straightforward approach aims to address.

C3. The paper needs to discuss and include more graph foundation models for experimental comparison.

C3-1. As mentioned in the paper, a classic approach to enabling LLMs for graph reasoning is transforming graphs into text descriptions. Although this approach may seem intuitively difficult to compare directly with the proposed embedding fusion method, the authors should still discuss and include it in experiments to demonstrate the superiority of their approach.

C3-2. Experimentally, the tasks considered in this paper do not cover some of the more challenging problems addressed by current graph foundation models. For example, graphWiz [1] focuses on classic graph problems such as bipartite, topology, triangle, Hamiltonian, and subgraph problems. The authors should discuss these tasks and consider including them.

C4. As discussed in C1, the paper’s technical approach combines parameter-efficient fine-tuning to drive LLMs for graph reasoning tasks. Therefore, the authors may need to discuss the efficiency of the proposed method to address concerns raised in C1.

Minor Concern:

C5. Lines 33-42 seem more like an introduction to the graph RAG task rather than the graph foundation model task.

**Reference**

[1] GraphWiz: An Instruction-Following Language Model for Graph Computational Problems, KDD 2024.

**Questions:**

Please check C1 to C4.

---

> ### Author Response · Authors · 2024-11-18
> **Rebuttal, Part 1**
>
> We thank the reviewer for their careful review of our paper. Responding to the concerns raised:
>
> ### **C1.) Is Text Enough?**
>
> While we appreciate that a text-only interface is indeed a benefit of other methods, we believe that alternative (and superior) methods for introducing graph structured data must exist.
>
> In this work, we are explicitly trying to demonstrate the benefits to graph reasoning that can come from treating graphs as a separate modality, in a similar way to how LLMs currently support vision and audio data.  While this does come at the cost of no longer being able to work with closed-source LLMs, we believe that our results demonstrate significant benefits for (re: Table 1, **>100%** performance on 5 tasks, and substantial improvement on the rest).
>
> We anticipate these strong results may motivate closed source LLMs to add support for graphs based on these results.
>
> ### **C2.) Contributions**
>
> The technical contribution of the GraphToken encoder is the extension of this 'fusion' approach to graphs including the comparison of it with simpler text approaches, extension to new applications (chemical property prediction), and studying the generalization of the representations produced.  *GraphToken was the first work to propose this fusion* (although there have been many concurrent (and subsequent works) since it was first released).
>
> The impact of the different degrees of freedom in encoding a graph in text [1] or even images [2] are well studied in prior work. [1] found that while certain text encoding methods perform better than others, there was broad weakness in LLMs ability to reason over graphs in this form.  [2] found that the image modality was insufficient for graphs (and generally did not add much over a text representation)
>
> ### **C3-1.) Comparisons**
>
> Comparison to text encoding methods is indeed an important baseline. In Table 1 in our paper we provide some comparisons between GraphToken and a few methods of text encoding with different prompting methods. We provide a brief description of the comparison in section 4.1. However due to length constraints we had to move a more in depth discussion of this part to the appendix.  Please see Appendix A.3.  We are happy to extend this discussion further!
>
> ### **C3-2.) GraphWiz**
>
> Thank you for calling our attention to GraphWiz! The GraphQA benchmark that we use covers some but not all of these tasks (in particular Hamiltonian and the subgraph matching problems look interesting to compare to).
>
> ### **C4.)  Efficiency**
>
> We will respond to this in its own comment for space
>
> ### **C5.) Minor comments**
>
> Thanks, for the suggestion!  We believe this paragraph to be related to our main goal since we seek to find the best way to augment a prompt with graph data (e.g. a knowledge graph might have fresher world knowledge than a LLM).  However, to your point, it does take a while to get there 🙂.   We will update this text to be relevant to the current draft.  (We do ask the reviewer to note that our task differs slightly from the ‘graph foundation model’ task as many define it.  Instead, we seek the best embedding of graph structure for its utilization in foundation models.)
>
> ### References:
>
> [1] Talk like a graph: Encoding graphs for large language models - https://arxiv.org/pdf/2310.04560
>
> [2] Which Modality should I use -- Text, Motif, or Image? : Understanding Graphs with Large Language Models - https://arxiv.org/abs/2311.09862

---

> ### Author Response · Authors · 2024-11-18
> **Rebuttal, Part 2**
>
> **C4.) Efficiency**
> =======
> Thanks for the question!   We see several angles to further the discussion.
>
> ### 1. **Training Speed**
>
> One way to illustrate the strength of our approach is by measuring the training speed of GraphToken compared to other PeFT baselines on the molecular property prediction task.  Please see the following table below:
>
> |  |  | Trainable parameters | BACE | BBBP | ClinTox\_2 |
> | :---- | :---- | ----- | ----- | ----- | ----- |
> | Model | Method |  | Training Hours | Training Hours | Training Hours |
> | Gemma-2B | Prompt tuning | 40960 | 1.55 | 2.14 | 1.49 |
> | Phi2-2.7B | Prompt tuning | 51200 | 1.63 | 2.27 | 1.71 |
> | Mistral-7B v0.3 | Prompt tuning | 81920 | 3.98 | 3.97 | 3.99 |
> | Gemma-2 2B | LoRA finetuning (input embed tuning) | 516096 | 0.30 | 0.47 | 0.28 |
> | Phi3.5 3.8B | LoRA finetuning (input embed tuning) | 70272 | 0.37 | 0.98 | 0.57 |
> | Mistral-7B v0.3 | LoRA finetuning (input embed tuning) | 73728 | 0.77 | 1.07 | 0.76 |
> | *Gemma-2 2B* | *P Tuning (1 token, 100k params)* | *97208* | *13.92* | *13.90* | *13.92* |
> | Phi3.5 3.8B | P Tuning (1 token, 100k params) | 129464 | 22.37 | 18.03 | 15.42 |
> | Mistral-7B v0.3 | P Tuning (1 token, 100k params) | 172472 | 23.95 | 23.89 | 23.87 |
> | **Gemma-2 2B** | **ChemGraphToken (MPNN)** | **299520** | **0.13** | **0.27** | **0.19** |
>
> From this table, we see that our method is not only the best performing method, but also the fastest!  The strongest performing baseline (P Tuning) takes **100x** more time to train than our method.
>
>
> ### 2. **Runtime Complexity**
>
> The design of the GNN encoder directly influences the runtime complexity for both training and inference in our method.  This is a significant strength of our proposed architecture, as GNN layers can have much better computational complexity than transformer layers.
>
> For instance, the complexity of a MPNN layer used in the above experiment is $O(|E|d)$ [3], while a transformer layer would require at least $O(|V|^2d)$ [4,5] to represent a graph ($d$ is the hidden dimension size).  In sparse graphs, $|E| << |V|^2$.  Most applications of graphs in the real world (molecules, social networks, etc) are sparse.
>
> ### 3. **Parameter Usage**
>
> While it is not the best measure of speed here (due to the architectural differences we just mentioned), we reference the parameter usage Table from Section 4.3.3, and the GraphQA experiments in Table 1.   In this we see that adding a very small number of “graph aware” parameters $O(100k)$ allowed a small model (PaLM-2 S) to outperform a very large model (PaLM-2 L).
>
> Quite frankly, **GraphToken allows a model that can run on a desktop PC to outperform a supercomputer**.  (We sincerely apologize that we can not be more specific about the exact parameters used by each model.)
>
> **References**
>
> [3] https://arxiv.org/pdf/2405.17311v1
>
> [4] https://arxiv.org/pdf/1706.03762
>
> [5] https://arxiv.org/pdf/2207.02505

---

> ### Author Response · Authors · 2024-11-22
> **GraphWiz Results**
>
> ### **C3-2.) Comparison to GraphWiz on harder graph problems**
>
> Thank you again for calling attention to the additional tasks presented in GraphWiz!
>
> To analyze GraphToken’s performance on some of these more challenging tasks, we focused on the Hamilton path graph problem proposed in GraphWiz, in which models predict whether a Hamiltonian path exists in the graph which visits each node exactly once.  Since GraphToken uses parameter efficient supervision, we compare the performance of GraphToken (with a GraphSAGE encoder) against the performance reported by GraphWiz for SFT:
>
> | Model | Parameters Tuned | Accuracy on Hamilton Path Task |
> | :---- | -----: | :---: |
> | Mistral 7B Naive SFT | 7,000,000,000 | 31.75 |
> | Mistral 7B GraphWiz | 7,000,000,000 | 26.50 |
> | Llama-2 7B Naive SFT | 7,000,000,000 | **69.00** |
> | Llama-2 7B GraphWiz | 7,000,000,000 | 52.25 |
> | Llama-2 13B Naive SFT | 13,000,000,000 | 59.75 |
> | Llama-2 13B GraphWiz | 13,000,000,000 | 59.00 |
> | Gemma-2 2B GraphToken | **338,688** | **68.15** |
>
> Again, we see very strong performance from GraphToken, matching the performance of the best results with a model that only requires tuning **338,688 parameters**!   This is fantastically more efficient (requiring only **0.0048384%** of the parameters tuned by the Llama-2 7B SFT baseline).
>
> Once again, we see that GraphToken’s ability to make smaller LLMs outperform much larger ones through encoding structure greatly aids in graph reasoning tasks.

---

> ### Author Response · Authors · 2024-11-26
> **Any additional concerns?**
>
> As we finish the first discussion phase, we'd like to call your attention to our results highlighting our efficiency, and a comparison on a NP-Complete task we performed at your request.
>
> Please let us know if you have any additional questions, and thanks again for your thoughtful review!

---

> > ### Author Response · Authors · 2024-12-02
> > **GraphWiz Subgraph Matching Results**
> >
> > ### **C3-2.) Comparison to GraphWiz on subgraph matching**
> >
> > In addition to our previous experiments on the Hamilton path graph problem, we have additionally run experiments on the **subgraph matching task**, the other NP-Hard task in GraphWiz. We compare GraphToken against all variants of GraphWiz on this task, despite GraphToken being a parameter-efficient method:
> >
> > | Model | Parameters Tuned | Accuracy on Subgraph Matching |
> > | :---- | ----- | ----- |
> > | Mistral 7B Naive SFT | 7,000,000,000 | 41.25 |
> > | Mistral 7B GraphWiz | 7,000,000,000 | 85.50 |
> > | Mistral 7B GraphWiz DPO | 7,000,000,000 | 48.50 |
> > | Llama-2 7B Naive SFT | 7,000,000,000 | 75.45 |
> > | Llama-2 7B GraphWiz | 7,000,000,000 | 82.25 |
> > | Llama-2 7B GraphWiz DPO | 7,000,000,000 | 77.25 |
> > | Llama-2 13B Naive SFT | 13,000,000,000 | 54.75 |
> > | Llama-2 13B GraphWiz | 13,000,000,000 | 81.50 |
> > | Llama-2 13B GraphWiz DPO | 13,000,000,000 | 77.00 |
> > | Gemma-2 2B GraphToken | **371,712** | **87.90** |
> >
> > We again see very strong performance from GraphToken, exceeding all GraphWiz variants while tuning a fraction of the parameters and using a smaller LLM base model. This demonstrates that GraphToken performs better and is much more efficient (0.0053102% of the tunable parameters of the Llama-2 SFT baseline) than other methods.
> >
> > We hope this provides further evidence of GraphToken’s ability to encode structure for LLMs in graph reasoning tasks.

---

### Author Response · Authors · 2024-11-25
**Thanks**

To all reviewers,

We thank you for your detailed reviews.  We believe our response has generated substantial evidence to further support our paper’s inclusion at ICLR.

### **New Revision Coming**
We are in the process of updating the paper to include experimental results generated during this discussion.  Specifically, we are adding  the following analyses:

-  **Multi-task Generalization** (requested by ***FJUo***) has been added to the Appendix.
     - *tl;dr*: **GraphToken representations can generalize to unseen graph tasks!**

-  **Runtime Analysis** (requested by ***Zq2s***, ***AQrB***) of PeFT methods has been added to the Appendix.
     - *tl;dr*: **GraphToken is faster than other PeFT methods and achieves strong results!**

-  **The Effect of Graph Structure** (requested by ***Zq2s***) on GraphToken performance
     - *tl;dr*: **GraphToken generalizes to different graph structures well!**

We will also include additional citations and references of concurrent work at the request of the reviewers, in addition to other minor changes.   However the primary context of the paper (and our contributions) are relatively unchanged.

Thanks again,

Authors

---

### Meta-Review · Area_Chair_wCiC · 2024-12-21

**Metareview:**

The paper introduces GraphToken, a parameter-efficient method for encoding structured graph data to enhance the graph reasoning capabilities of LLMs. The authors leverage GNNs to encode graph structures into tokens that are then integrated with text embeddings for input into LLMs. While the experimental results demonstrate significant improvements on certain graph reasoning tasks, the submission falls short in several critical areas that impede its suitability for acceptance at ICLR.

1. Limited novelty: The core mechanism of embedding fusion using GNNs and text encoders is not sufficiently differentiated from existing approaches. The paper does not clearly articulate what makes GraphToken technically superior or distinct beyond parameter efficiency. There are many existing works encoding graphs to LLM space, like the four papers listed by reviewer FJUo. After checking the submission's references, the LLaGA paper is wrong referenced under GOFA, and the GOFA paper is even not mentioned, reflecting the carelessness in discussing related work.

2. Weak experiments: The paper lacks comprehensive comparisons with state-of-the-art graph foundation models such as GraphWiz and others cited by reviewer FJUo. This omission makes it difficult to assess the true efficacy of GraphToken relative to leading methods. The performance improvements appear to stem more from the strengths of the GNN encoder rather than the seamless integration with LLMs. This raises questions about the actual contribution of the GraphToken framework in enhancing LLM capabilities. Although the authors provided new results of only using their GNN encoder in response to FJUo, I found the results actually indicated in most tasks GNN only is superior than GraphToken (GNN + LLM). This further raises concerns on the necessity and motivation for integrating LLMs in these tasks.

3. Limited scope: Reviewer FJUo highlighted that the integration with LLMs does not convincingly demonstrate enhanced in-context learning or generalizability. The experiments remain largely supervised-task-oriented without showcasing the broader applicability of LLMs in zero-shot or few-shot settings. The method's reliance on encoding graphs into single tokens also poses scalability issues for larger or more complex graphs. The paper does not adequately address how GraphToken performs as graph size increases beyond the benchmarks used.

**Additional Comments On Reviewer Discussion:**

Although the authors made efforts to address some concerns raised during the review process, critical issues such as comprehensive baseline comparisons and in-context learning evaluations remain inadequately resolved. Further, the GNN only results strengthens the concerns on using GNN + LLM on such tasks.

---

### Decision · Program_Chairs · 2025-01-22

Reject